# Hydroclimate shapes photosynthetic sensitivity to cloud cover across global terrestrial ecosystems

Hao Luo [1] ✉, Ana Bastos [2], Markus Reichstein[3], Gregory Duveiller [3], Jan Kretzschmar [1] & Johannes Quaas [1,4]

Vegetation photosynthesis primarily depends on surface energy and water availability, both of which are simultaneously regulated by clouds through radiation and precipitation, respectively. However, the net impact of cloud-induced changes in surface solar radiation and precipitation on photosynthesis remains elusive. Here, using observational- and model-based datasets spanning the past few decades, we show that, consistently across scales from site-level eddy covariance measurements to global-scale gridded datasets, the sensitivity of photosynthesis to cloud cover is spatially shaped by the hydroclimate, as quantified by the humidity index (mean annual precipitation-to-evapotranspiration ratio). Specifically, we find that in water-limited arid regions, clouds promote photosynthesis through increased precipitation, with a delayed effect typically within one month, whereas in energy-limited humid regions, they inhibit photosynthesis almost instantaneously by blocking sunlight. An annual scale spatially resolved sensitivity metric of photosynthesis to cloud cover is further examined to estimate potential changes in vegetation productivity driven by clouds. The findings indicate that, under a warming climate, particularly in the Coupled Model Intercomparison Project Phase 6 "ssp585" scenario (2015–2099), gross primary productivity is projected to decline in arid regions and increase in humid regions due to changes in cloud cover, suggesting an exacerbation of regional disparities in ecosystem functions.

Vegetation is an integral component of the Earth's climate system, with photosynthesis driving the largest carbon flux over global land surfaces[1] and the balance of photosynthesis, respiration, and disturbance fluxes resulting in a net sequestration of approximately a quarter of annual anthropogenic carbon dioxide ($CO_2$) emissions[2]. Vegetation functioning and dynamics are influenced by various environmental factors, including precipitation, solar radiation, temperature, humidity, and $CO_2$ concentration[3–8]. However, spatial patterns of the vegetation carbon uptake are primarily governed by the balance between energy availability and water limitation[9,10]. In energy-limited environments, such as high-latitude ecosystems and tropical rainforests, reduced photosynthetically active radiation (PAR) and low temperature constrain photosynthetic activity[11–15]. Conversely, in water-limited regions, such as arid and semi-arid landscapes, precipitation and soil moisture availability impose major constraints on carbon assimilation, even when sunlight and temperature conditions are favorable[16–20]. Therefore, recognizing how these limitations interact across spatial and temporal scales is crucial for studying vegetation responses to environmental conditions and future climate change[5,10].

[1]Leipzig Institute for Meteorology, Leipzig University, Leipzig, Germany. [2]Institute for Earth System Science and Remote Sensing, Leipzig University, Leipzig, Germany. [3]Max Planck Institute for Biogeochemistry, Jena, Germany. [4]German Centre for Integrative Biodiversity Research (iDiv) Halle-Jena-Leipzig, Leipzig, Germany. ✉e-mail: hao.luo@uni-leipzig.de

The environmental variables that govern vegetation dynamics have been typically examined in previous studies either independently[21–24] or in combination[5,6,10,25–30]. Yet clouds, which simultaneously regulate surface radiative flux and precipitation[31–33], have received little attention regarding their complex impacts on vegetation photosynthesis[5,34]. In terms of radiative effects, clouds influence the surface energy budget both by reflecting sunlight back into space (albedo effect) and trapping terrestrial radiation (greenhouse effect)[35]. Generally, clouds contribute to surface cooling across most regions due to their dominant albedo effect, except over some bright areas and for high solar zenith angles (e.g., Greenland, the Arctic, and Antarctica)[33,36]. Thus, variations in clouds and associated fluctuations in surface radiation can impact photosynthetic efficiency by altering light availability and temperature conditions. In terms of precipitation, the presence of clouds is a necessary condition for rainfall and snowfall, with cloud formation and dynamics governing the amount, timing, and frequency of precipitation events[37,38]. These precipitation properties play crucial roles in determining soil moisture levels and, consequently, the water availability for photosynthesis[29].

Cloud fraction (CF), a key macro-characteristic of clouds that quantifies their horizontal coverage, is closely linked to both surface radiation and precipitation, although cloud thickness, type, and altitude can modulate the CF-radiation-precipitation relationship to some extent[39–41]. For instance, thick cumulonimbus clouds are optically opaque and more likely to produce heavy rainfall, whereas high-altitude cirrus clouds, which are relatively optically thin, allow more sunlight to penetrate and have a low precipitation frequency. In a warming climate, CF is expected to decline due to changes in thermodynamic structure and moisture conditions as concluded from both observations and simulations, with the most pronounced reductions found in low-level clouds[33,42,43]. As low-level clouds play critical roles in reflecting incoming solar radiation and regulating surface precipitation, their decline may alter environmental conditions that drive vegetation dynamics. Hence, given their dual role in modulating both energy and water availability, understanding the effects of CF on photosynthesis is essential for a more comprehensive assessment of ecosystem carbon dynamics and vegetation productivity under a changing climate.

In this study, using CF as a combined proxy for radiation and precipitation limitations, we determine to what degree global vegetation dynamics are sensitive to variations in CF. We use gross primary productivity (GPP) data from both in-situ eddy-covariance measurements from the FLUXNET network[44] and upscaling gridded FLUXCOM-RS dataset[45] as benchmarks for analysis (see "Methods" for more details). Additional proxies of photosynthesis from various sources are also used to validate the robustness of our results. CF data are sourced from moderate resolution imaging spectroradiometer (MODIS)[46] observations and the European Centre for Medium-range Weather Forecasts (ECMWF) fifth generation reanalysis (ERA5)[47]. Hydroclimatic conditions are classified as either water- or energy-limited based on the humidity index (HI), which is calculated as the long-term mean annual precipitation amount (PA) divided by potential evapotranspiration (PET)[9]. With these datasets, we examine how the sensitivity of vegetation photosynthesis to CF varies across different hydroclimatic conditions and cloud regimes. We further explore the respective roles of cloud-mediated radiative and precipitative effects in regulating vegetation responses. Finally, we assess how projected changes in CF under global warming reshape the spatial patterns of vegetation dynamics.

## Results and discussion
### Spatial patterns of photosynthetic sensitivity to CF are shaped by the hydroclimate
Estimates of the GPP sensitivity to CF (Eq. 4 and "Methods"), derived from both in-situ measurements and global gridded datasets, reveal a clear spatial dependence on hydroclimate (Fig. 1). The investigations are based on daily estimates of GPP from eddy-covariance measurements from FLUXNET[44] combined with CF data from the ERA5 reanalysis[47] at the nearest grid point, and 8-daily GPP data from FLUXCOM-RS[45] alongside CF data from MODIS/Terra observations[46]. All data are standardized by removing long-term trends and seasonal cycles (Eqs. 1–3 and "Methods") prior to calculating the sensitivities, and are subsequently marked with an asterisk (i.e., GPP* and CF*) to distinguish them from the original data. We define the HI ("Methods") to quantify the hydroclimate, where smaller values indicate more arid regions, while larger values represent more humid regions. Both the inter-site comparisons and the global map evidently show a less negative sensitivity of GPP*-to-CF* as HI decreases. As the focus shifts from humid to arid regions, the strong negative sensitivity transforms into weak positive values, with the significance of the sensitivity substantially reduced in arid regions. However, on a global average, negative GPP*-to-CF* sensitivity is diagnosed. Further sensitivity investigations, including examination of different vegetation types and comparison with independent datasets, are discussed in the Supplementary Materials.

In addition to the 8-daily temporal resolution analysis, the global scale estimation of GPP*-to-CF* sensitivity is also performed using monthly average FLUXCOM-RS GPP data and MODIS/Terra CF data to facilitate comparisons across different temporal scales (Supplementary Fig. 1). The results indicate that the GPP*-to-CF* sensitivity derived from the monthly averages remains spatially shaped by HI. The primary difference is that the sensitivity estimated from the monthly data is generally higher than that from the 8-daily analysis, particularly showing a clearer positive sensitivity in arid regions, with the underlying reasons discussed from the perspective of cumulative precipitation effects in the subsequent sections. This comparison across different temporal scales demonstrates that GPP*-to-CF* sensitivity analysis can also be reliably conducted using data at the monthly scale.

Therefore, we further conduct a causal analysis based on monthly GPP estimates from 20 dynamic global vegetation models included in the TRENDYv12 project, as these models are offline simulations that do not account for vegetation-climate feedback[48]. Our analysis uses outputs from simulation scenario S2 (scenario defined in the TRENDY project, more details in the "Methods" section), which incorporates temporal variations in climate and $CO_2$ concentrations, while keeping land cover stable at preindustrial levels to exclude the potential confounding effects from land use changes. We consider the multi-model ensemble mean. It should be noted that the TRENDY models do not directly use cloud as a forcing, but rather radiation, precipitation, and specific humidity, which are, of course, closely related to changes in CF. In addition, the effect of diffuse radiation is included in some models[48], but changes in diffuse fraction due to clouds or aerosols are not included in the forcing[49]. The GPP*-to-CF* sensitivity estimated using the TRENDY modeled GPP and MODIS/Terra observed CF exhibits spatial patterns similar to those observational results discussed above, with sensitivity decreasing as HI increases (Supplementary Fig. 2). The MODIS/Terra CF is used only for diagnostic correlation with TRENDY GPP, not as a model input. This suggests that the sensitivity is primarily explained by cloud-induced changes in environmental factors that affect vegetation dynamics, rather than vegetation dynamics influencing clouds in turn. While previous studies have suggested that vegetation changes can influence surface turbulent fluxes and thus affect cloud formation, these findings are primarily observed or simulated in the context of large-scale substantial land cover changes, such as deforestation and afforestation[50,51]. Furthermore, the causality is also demonstrated through a data-driven time-lagged sensitivity analysis ("Methods"). By applying an 8- or 16-day lag (backward shift) or lead (forward shift) to the 8-daily CF* time series, we find that the sensitivity of GPP* to lagged CF* aligns more closely with their instantaneous sensitivity (Supplementary Fig. 3). This indicates that changes

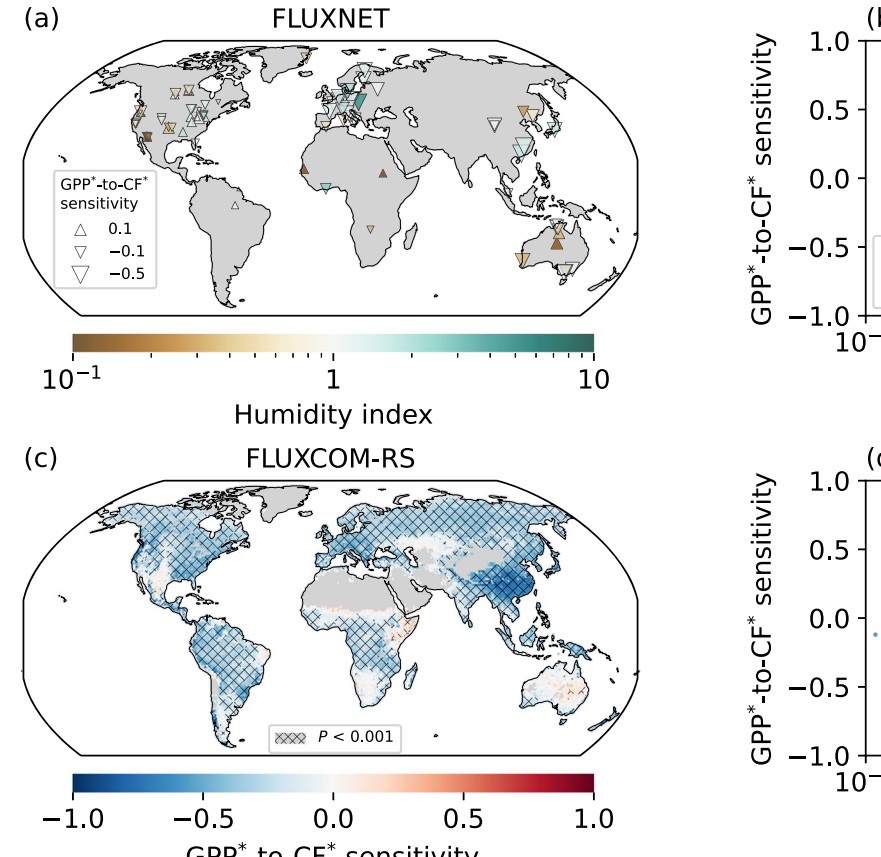
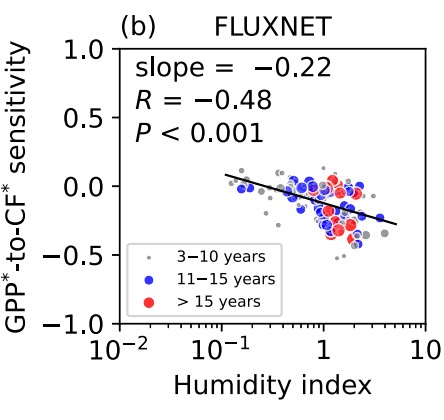
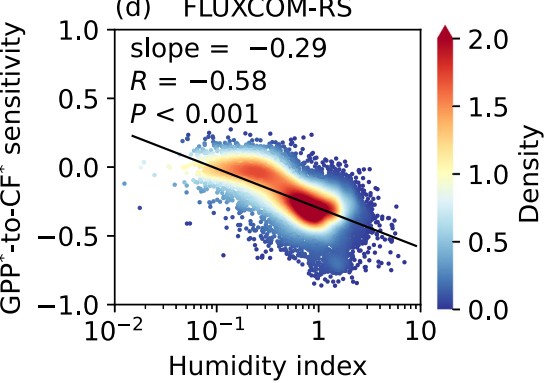

**Fig. 1 | Humidity index (HI) spatially shapes the sensitivity of gross primary productivity (GPP) to cloud fraction (CF) across global ecosystems. a** GPP*-to-CF* sensitivity (Eq. 4 and "Methods") derived from daily GPP measurements at FLUXNET and CF matched from the nearest grid of European Centre for Medium-Range Weather Forecasts (ECMWF) fifth-generation reanalysis (ERA5) data. The size of the triangles represents the magnitude of the values, with upward triangles indicating positive values and downward triangles indicating negative values. The triangles in the legend serve as reference scales for the values of 0.1, −0.1, and −0.5. The color filling inside each triangle corresponds to the average HI observed during the measurement period at each site. **b** Scatter plot showing the variation in GPP*-to-CF* sensitivity as a function of HI across sites in (**a**) on a logarithmic scale. GPP*-to-CF* sensitivity is modeled as a linear regression of the logarithmic HI, with the texts displaying the slope of the linear fit, correlation coefficient (R), and P-value from a Student's t test. The linear regression is represented by the black line. The size of each scatter point represents the length of the data record at each site, with larger points indicating longer measurement periods. Different colors are used to distinguish between different ranges of measurement durations. **c** Map of GPP*-to-CF* sensitivity derived from 8-daily GPP from FLUXCOM-RS and CF from the moderate resolution imaging spectroradiometer (MODIS) onboard Terra. Non-vegetated areas are masked ("Methods"). The cross-hatched areas represent regions where the P-value from a Student's t test is less than 0.001 in (**b**) but with density plot across grids in (**c**), and HI calculated based on data from the Climatic Research Unit time-series version 4.08 (CRU TS v4.08). The color bar shows the Gaussian kernel density estimate. The analysis in (**c**, **d**) is based on the periods between 2001 and 2020. The asterisk (*) indicates that the data have been standardized by removing long-term trends and seasonal cycles (Eqs. 1–3 and "Methods").

in CF have a stronger influence on future GPP, whereas the impact of GPP changes on future CF is relatively weak. In arid regions, the stronger positive impact of lagged CF on GPP is mainly driven by delayed precipitation effects, with the effective response time further quantified in the section "perspective of temporal scales". In summary, the conclusions from both the causal analysis based on offline simulations and the time-lagged analysis indicate that the calculated GPP*-to-CF* sensitivity is predominantly driven by the one-way influence of CF on GPP.

**Potential mechanisms explained by competing cloud radiative and precipitative effects**

Clouds play two major roles in the Earth-atmosphere system: firstly, by modulating the radiative budget through their interactions with radiation[33,52], and secondly, by regulating the water cycle through generating precipitation[53,54]. Both processes impact the energy and water availability, which is essential for photosynthesis[3], yet to some extent, their effects counteract each other. Specifically, while clouds reduce the energy available for photosynthesis by blocking solar shortwave radiation, the precipitation they generate enhances the water supply required for this process. Consequently, in specific regions, the sensitivity of photosynthesis to CF depends on which factor dominates. Thus, we discuss the potential mechanisms from the perspective of both cloud radiative and precipitative effects.

Similar to the analysis of GPP*-to-CF* sensitivity, we calculate the GPP*-to-PAR* sensitivity and GPP*-to-PA* sensitivity separately. For FLUXNET, we use surface incoming shortwave radiation (SWin) as a proxy for PAR, as they are approximately linearly correlated[9]. Globally, the GPP*-to-PAR* sensitivity is predominantly positive, gradually decreasing and even turning negative as HI decreases, with the significance (P-value) being less pronounced in negative regions (Supplementary Fig. 4). Additional analysis of the direct and diffuse components of PAR reveals that the direct component accounts for most of the effect of total PAR on GPP, whereas the contribution of diffuse PAR is relatively limited (Supplementary Fig. 5). While diffuse PAR has been recognized as a positive driver of GPP in previous studies[55,56], its influence remains weaker than that of direct PAR at the global scale[57]. In contrast, the GPP*-to-PA* sensitivity is primarily

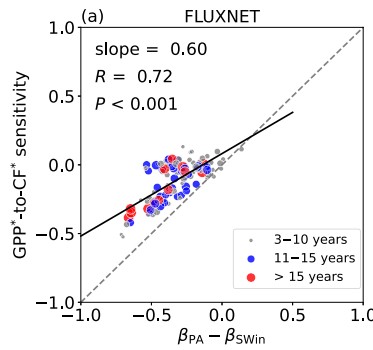
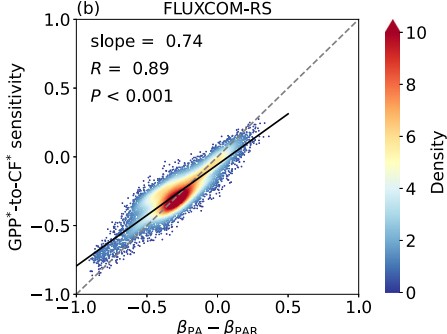
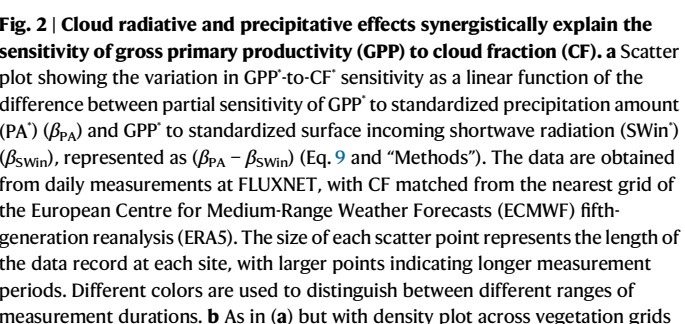

**Fig. 2 | Cloud radiative and precipitative effects synergistically explain the sensitivity of gross primary productivity (GPP) to cloud fraction (CF). a** Scatter plot showing the variation in GPP*-to-CF* sensitivity as a linear function of the difference between partial sensitivity of GPP* to standardized precipitation amount (PA*) ($\beta_{PA}$) and GPP* to standardized surface incoming shortwave radiation (SWin*) ($\beta_{SWin}$), represented as ($\beta_{PA} - \beta_{SWin}$) (Eq. 9 and "Methods"). The data are obtained from daily measurements at FLUXNET, with CF matched from the nearest grid of the European Centre for Medium-Range Weather Forecasts (ECMWF) fifth-generation reanalysis (ERA5). The size of each scatter point represents the length of the data record at each site, with larger points indicating longer measurement periods. Different colors are used to distinguish between different ranges of measurement durations. **b** As in (**a**) but with density plot across vegetation grids

using 8-daily GPP from FLUXCOM-RS, CF from the moderate resolution imaging spectroradiometer (MODIS) onboard Terra, PA from the Global Precipitation Measurement (GPM) Integrated Multi-satellitE Retrievals for GPM (IMERG), and photosynthetically active radiation (PAR) from the Clouds and the Earth's Radiant Energy System (CERES). The partial sensitivity of GPP* to PAR* ($\beta_{PAR}$) is used instead of $\beta_{SWin}$. The color bar shows the Gaussian kernel density estimate. The analysis in (**b**) is based on the periods between 2001 and 2020. In both (**a, b**), the linear regressions are represented by the black lines, with the texts displaying the slope of the linear fit, correlation coefficient ($R$), and $P$-value from a Student's $t$ test. The 1:1 line in each subplot is marked with a gray dotted line. The asterisk (*) indicates that the data have been standardized by removing long-term trends and seasonal cycles (Eqs. 1–3 and "Methods").

negative worldwide, but as HI decreases, its value gradually increases and ultimately becomes positive (Supplementary Fig. 6). Both sensitivities exhibit a clear linear correlation with changes in the logarithm of HI, and their spatial patterns align well with those found by previous studies regarding water and energy limitations on vegetation activity[3,5,9,26,29].

Furthermore, to quantify the relative contributions of precipitation and radiation to GPP, we apply a multiple linear regression with PAR* and PA* as two regressors and GPP* as the response variable (Eq. 9 and "Methods"). Since the data are standardized, the values of the two partial sensitivities can be directly compared. Hence, the difference between the partial sensitivity of GPP* to PA* ($\beta_{PA}$) and GPP* to PAR* ($\beta_{PAR}$) (i.e., $\beta_{PA} - \beta_{PAR}$) serves as a proxy of the relative importance of precipitation and radiation on GPP. The global patterns of $\beta_{PA} - \beta_{PAR}$ exhibit the same linear response to the logarithm of HI as the GPP*-to-CF* sensitivity, showing positive values in more arid regions, but negative ones in more humid regions (Supplementary Fig. 7). Further comparisons of $\beta_{PA} - \beta_{PAR}$ and GPP*-to-CF* sensitivity in the inter-site or inter-grid scatter plots show a close alignment along the 1:1 line (Fig. 2), indicating that the sensitivity of GPP to CF is effectively explained by the balance between its sensitivity to precipitation and radiation. In water-limited arid regions with low HI, clouds promote photosynthesis by generating precipitation to enhance water supply, while in energy-limited humid regions with high HI, clouds inhibit photosynthesis by blocking sunlight to limit energy availability. The disparity in the dominant mechanisms of cloud effects on vegetation across regions explains why the spatial sensitivities of photosynthesis to CF are shaped by the hydrological climate.

## Impacts of cloud types and properties

The impacts of clouds on radiation and precipitation are, to some extent, influenced by cloud types[33,58,59]. In general, ice clouds in the upper troposphere are nearly transparent to sunlight and produce minimal precipitation, while liquid clouds reflect a large amount of solar radiation and generate substantial precipitation. The perturbations of clouds on solar radiation and precipitation are also closely related to cloud optical thickness (COT), with deep convective clouds characterized by larger COT exhibiting higher albedo and more intense precipitation[32,60,61]. Accordingly, classifying cloud types allows

us to gain a more comprehensive understanding of how vegetation dynamics respond to varying cloud regimes, thereby revealing deeper insights into the underlying mechanisms.

Here, we firstly distinguish between single-layer ice cloud fraction (CFice) and single-layer liquid cloud fraction (CFliquid) retrieved from MODIS and separately evaluate their impacts on GPP. Both inter-site (Supplementary Fig. 8c–f) and global scale (Supplementary Fig. 9) analysis indicate that GPP*-to-CF* sensitivity is more strongly influenced by liquid clouds, while the contribution of ice clouds is relatively weak. This result aligns with the known cloud regimes-radiation-precipitation relationship. Moreover, the grid-mean COT ($\overline{COT}$), defined as $\overline{COT} = CF \times \overline{COT}_{cloudy}$, where $\overline{COT}_{cloudy}$ represents the average COT over cloudy pixels only, is applied to study the sensitivity of GPP to cloud vertical extension. The GPP*-to-$\overline{COT}$* sensitivity shown in Supplementary Fig. 10 exhibits nearly identical spatial patterns to GPP*-to-CF* sensitivity, following a linear decreasing function of the logarithmic HI. This suggests that the influence of clouds on vegetation dynamics remains consistent across both horizontal coverage and vertical extension. Increased horizontal coverage and deeper vertical extension both enhance solar radiation reflection and precipitation formation, thereby shaping the spatial patterns of vegetation dynamics. These two processes operate simultaneously, as larger CF is usually associated with deeper clouds, particularly in convective cloud regimes[32].

At the same time, we assess the partial sensitivity of GPP to nine cloud types and their relative contributions (Eqs. 5 and 6; "Methods") using 8-daily joint histograms of COT and CTP data from MODIS/Terra[62], where we categorize the cloud types based on the International Satellite Cloud Climatology Project (ISCCP) classification scheme[61]. High surface albedo over bright regions complicates cloud detection from the passive satellite data, resulting in substantial data gaps in the joint histograms. Therefore, the analysis preferentially covers humid regions where data qualities are more reliable (Supplementary Figs. 11 and 12). The results reveal that clouds with COT exceeding 3.6 generally contribute positively to the sensitivity of GPP to total CF, with thicker clouds contributing more due to enhanced solar radiation attenuation and higher precipitation potential. Conversely, in most humid regions, clouds with COT below 3.6, especially cirrus clouds with negligible precipitation, show a negative relative

contribution. This indicates that these translucent clouds impose limited constraints on direct radiation, while the associated increase in diffuse radiation enhances GPP[11,63]. However, additional analysis combining COT and total CF data shows that these thin clouds with COT below 3.6 are primarily distributed in arid regions (Supplementary Fig. 13a). Consequently, their absolute negative contribution to the sensitivity of GPP to total CF in humid regions is minimal. Moreover, the data gaps in the joint histograms of COT and CTP over arid regions are partially compensated by the results presented in Supplementary Fig. 13, revealing that clouds of different optical thicknesses tend to positively affect GPP in these regions.

In summary, while the analysis based on cloud regimes reveals certain differences in the impacts of individual cloud types on vegetation dynamics, the overall patterns suggest that total CF captures the essential cloud-related effects. These findings support the use of total CF as a sufficient and robust indicator for characterizing cloud properties in studies of vegetation sensitivity to clouds.

## Shifts in CF-driven GPP from arid to humid regions in a warming climate

The annual scale GPP-to-CF sensitivity is further estimated using a multiple linear regression model based on year-to-year variations to examine the long-term changes in GPP driven by CF (Eq. 7 and "Methods"). Unlike GPP*-to-CF* sensitivity estimation, which focuses only on obtaining spatial signals, our objective here is to derive actual annual scale values of GPP-to-CF sensitivity, so we incorporate additional environmental constraints, such as 2 m temperature (T2m) and specific humidity ($q$) (Eq. 7 and "Methods"). However, the issue of multi-collinearity among these regressors has to be addressed through validations of the variance inflation factor (VIF) ("Methods"). Since vapor pressure deficit (VPD) is a function of temperature and humidity[64,65], the latter also serves as a constraint. The total variance explained ($R^2$) shown in Supplementary Fig. 14a indicates that the model has strong overall explanatory power, although its performance is relatively weaker in low-latitude regions. Among the predictors, CF plays a prominent role in the regression model, as reflected by its incremental variance explained ("Methods"), which is higher or comparable in magnitude to that of T2m and $q$ (Supplementary Fig. 14b–d). The results demonstrate that the spatial patterns of annual scale GPP-to-CF sensitivity are, as for the standardized values, shaped by hydroclimate, with a linear decreasing function of the logarithmic HI (Fig. 3). The annual scale GPP-to-CF sensitivity exhibits high spatial consistency with the monthly scale GPP*-to-CF* sensitivity, with the proportion of spatial pixels with consistent sensitivity signs exceeding 70%. The remaining approximately 30% of pixels exhibit differences mainly in the transition regions between positive and negative values.

Before calculating the temporal variations in CF-driven GPP, we assess the long-term changes and trends in CF over the historical period from 1984 using extrapolated in situ observations: the Climatic Research Unit time-series version 4.08 (CRU TS v4.08), satellite observations: the ISCCP high-resolution global monthly product (ISCCP-HGM), reanalysis: the ERA5, and climate models: the sixth phase coupled model intercomparison project (CMIP6) 39 models ensemble mean, as well as future projections with the strong-warming scenario (ssp585) from CMIP6 between 2015 and 2099 ("Methods"). A consistent finding across all datasets and time periods is that the average CF over land vegetation areas declines under a warming climate (Supplementary Figs. 15 and 16), aligning with the conclusions of previous studies[42,43,66]. Besides, historical records show that the Earth has undergone periods of global dimming (1950s–1980s) and brightening (since the late 1980s), primarily driven by variations in clouds and aerosols[67,68]. Our analysis, based on observations during the brightening period, is consistent with this large-scale trend, as the observed decline in global average CF corresponds to reduced cloud albedo and increased surface solar radiation. However, CF trends are

not monotonic across all regions, and the observed brightening signal also reflects the reduction in aerosol levels in many regions[66]. The significant ($P < 0.05$) decreasing trends over the historical period (1984–2014) are less spatially extensive, with spatial patterns that are also not entirely consistent across different datasets. However, we emphasize that our primary focus is on assessing long-term trends in CF and the resulting potential changes in GPP under a warming climate. Our findings remain robust across different cloud climate records with similar global average declining CF trends, spanning both past and future periods, even when accounting for spatial disparities among datasets.

Consequently, the widespread declining trends in CF under a warming climate suggest that CF potentially plays a notable role in driving changes in GPP. We empirically estimate the impacts of changes and trends in CF on GPP based on the annual scale GPP-to-CF sensitivity (Eqs. 11 and 12; "Methods"). Although CF trends are uncertain across different datasets, estimates from each cloud climate record consistently show that CF-driven GPP trends typically decrease in arid regions but increase in humid regions, with minimal changes on a global average (Fig. 4). Thus, our findings demonstrate a spatial shift in GPP from arid to humid regions due to changes in CF under a warming climate (Fig. 4 and Supplementary Fig. 17). This indicates that as GPP is already low in arid regions, further shifts towards humid regions would lead to greater regional imbalances in carbon sequestration capacity, potentially exacerbating the challenge of mitigating climate change. By and large, such shifts in CF-driven GPP intensify regional disparities in ecosystem functions and lead to larger-scale changes in global carbon and water cycles under a warming climate.

It should be noted that clouds remain one of the most uncertain components in climate models because their sub-grid processes are not explicitly resolved and rely on imperfect parameterizations[69,70]. These limitations may contribute to uncertainties in projected CF changes and, consequently, in the estimated vegetation responses. Moreover, under a warming climate, future changes in clouds, radiation, precipitation, and vegetation are expected to co-evolve with adjustments in the hydrological cycle. Here, we highlight several potential processes that respond to climate change: (i) the atmosphere's water-holding capacity increases by about 7% °C$^{-1}$, while global precipitation rises more slowly by roughly 2–4% °C$^{-1}$ due to energetic constraints[71]; (ii) cloud vertical structure is projected to change, featuring a faster decline in liquid clouds than in ice clouds over land, thereby influencing cloud radiative effects[33]; (iii) GPP responds negatively to increasing VPD while exhibiting a nonlinear response to soil moisture[18]; (iv) thermal acclimation buffers photosynthesis against warming, reducing increases in photosynthesis under cold climates and limiting declines in photosynthesis under hot climates[72]. Collectively, these adjustments imply that cloud and vegetation characteristics will vary according to regional hydroclimate, potentially altering the radiative and precipitative impacts on GPP and introducing uncertainties in the estimation of CF-driven GPP changes shown in Fig. 4. Therefore, as our projections assume that the GPP-to-CF sensitivity inferred from the recent past (2001–2020) remains constant throughout the 21st century, future studies should aim to quantify the potential uncertainties under changing $CO_2$, VPD, hydrological cycle, and vegetation acclimation.

## Perspective of temporal scales

Evidence of the relative roles of cloud radiative and precipitative effects on vegetation dynamics is also examined from the perspective of the temporal scale. On the one hand, investigations within the meteorological active seasons ("Methods") reveal spatial patterns of GPP*-to-CF* sensitivity similar to those observed in the full-year analysis, with notably more negative values in boreal regions (Supplementary Fig. 18). In these energy-limited humid regions, the role of cloud

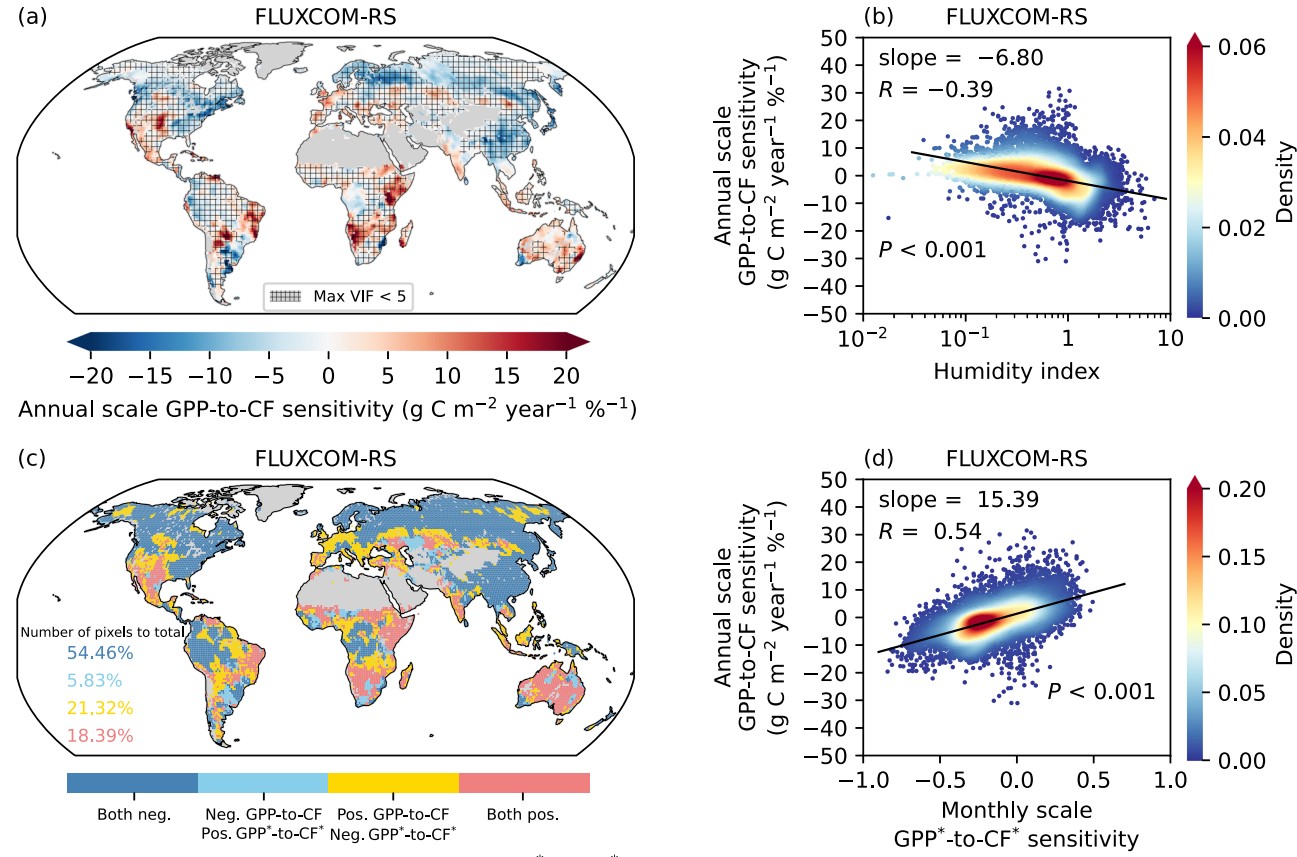

**Fig. 3 | Estimated annual scale sensitivity of gross primary productivity (GPP) to cloud fraction (CF). a** Annual GPP-to-CF sensitivity derived from multiple linear regression on year-to-year variation data (Eq. 7 and "Methods"). This estimate is based on the annual GPP from FLUXCOM-RS, CF from the moderate resolution imaging spectroradiometer (MODIS) onboard Terra, as well as 2 m temperature (T2m) and specific humidity (q) from the Climatic Research Unit Japanese Reanalysis (CRU JRA). The grid-hatched areas indicate regions where the maximum variance inflation factor (VIF) is less than 5. **b** Density plot showing the variation in annual GPP-to-CF sensitivity as a function of logarithmic humidity index (HI) across grids in (**a**). **c** A comparison between annual scale GPP-to-CF sensitivity and monthly scale GPP*-to-CF* sensitivity, with different colors indicating the consistency or inconsistency of the two symbols, and the proportion of pixels corresponding to each color labeled. GPP data are from FLUXCOM-RS, and CF data are from MODIS/Terra. **d** Density plot showing the comparison between annual scale GPP-to-CF and monthly scale GPP*-to-CF* sensitivity across grids in (**c**). Non-vegetated areas are masked in (**a**, **c**) ("Methods"). In both (**b**, **d**), the linear regressions are represented by the black lines, with the texts displaying the slope of the linear fit, correlation coefficient (R), and P-value from a Student's t test. The color bars show the Gaussian kernel density estimates. The analysis is based on the periods between 2001 and 2020. The asterisk (*) indicates that the data have been standardized by removing long-term trends and seasonal cycles (Eqs. 1–3 and "Methods").

radiative effects is diminished in the full-year analysis, as changes in radiation barely affect vegetation dynamics during the non-growing seasons—a pattern evident in boreal regions, such as Europe during winter, where the GPP*-to-CF* sensitivity shifts to positive (Supplementary Fig. 19g). On the other hand, the impact of precipitation on vegetation is mediated by changes in soil moisture, which involves a time delay and cumulative effects[73]. This explains the issue that GPP*-to-CF* sensitivity calculated using monthly-scale data shows a global enhancement compared to the 8-daily analysis (Supplementary Fig. 1c). This enhancement is due to the accumulative effects of precipitation on soil moisture, and consequently, on vegetation dynamics at the monthly scale. Additionally, the greater GPP*-to-CF* sensitivity observed in water-limited arid regions when CF is lagged in the time-lagged sensitivity analysis is attributed to the delayed effect of precipitation (Supplementary Fig. 3a, c). Further quantitative analysis (Eq. 10 and "Methods") reveals that, in regions where the time-lagged response of GPP* to PA* is statistically significant (P < 0.001), over 90% of pixels globally show a delayed effect for less than 1 month, with notable delays found in water-limited arid regions (Supplementary Fig. 20). Therefore, we conclude that the cloud radiative effects on vegetation are virtually instantaneous, while the effects of precipitation exhibit a time delay, typically within 1 month.

## Performance of CF in regressing GPP

We compare the root mean squared error (RMSE) from a univariate linear regression of GPP* on CF* with the RMSE from a multiple linear regression of GPP* on PA* and PAR*. The results show only a small difference, with a correlation coefficient of 0.88 and the RMSE of GPP* predicted by CF* being slightly higher (Supplementary Fig. 21), indicating that CF as a single variable performs similarly to a multiple regression model using PA and PAR. In addition, the direct correlation analysis shows statistically strong positive correlations between PA* and CF* and strong negative correlations between PAR* and CF* at the global scale (Supplementary Fig. 22), further supporting the use of total CF as an effective combined proxy for precipitation and radiation in this study. However, our statistical investigation of the CF*–PA* relationship is based on 8-daily averages at the global scale. At instantaneous or daily timescales, this relationship is expected to be weaker and more nonlinear[74], which should be taken into account when interpreting short-term dynamics.

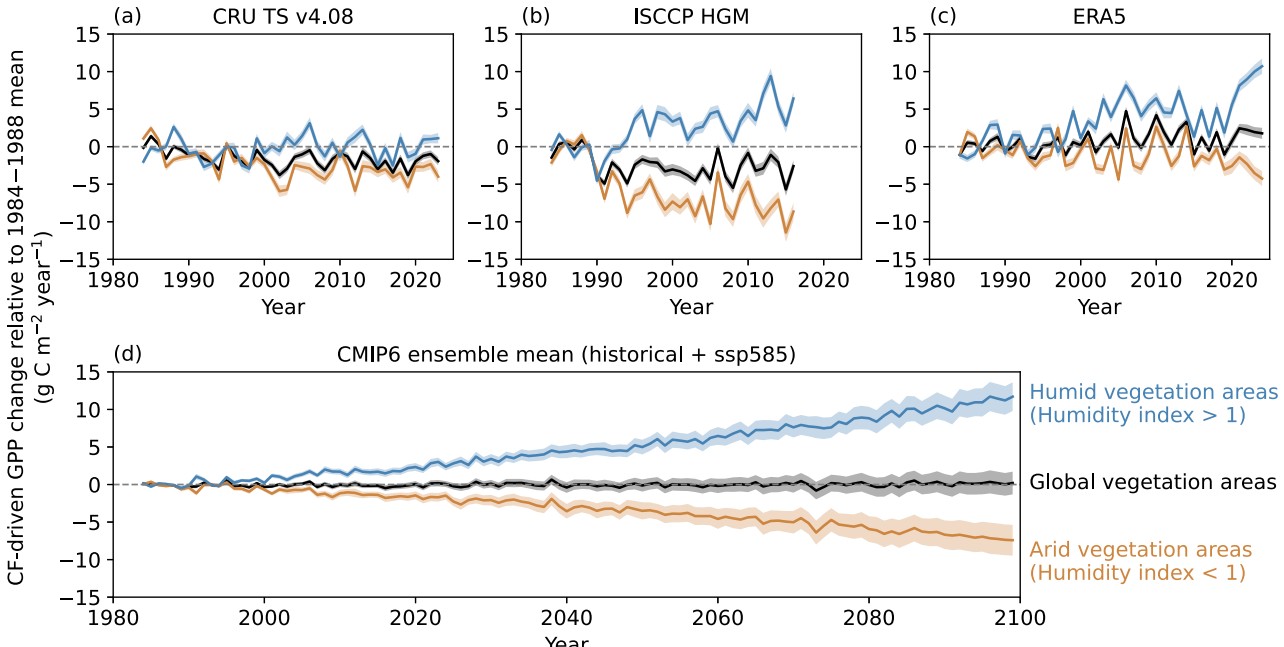

**Fig. 4 | Shifts in gross primary productivity (GPP) from arid to humid regions driven by cloud fraction (CF) changes in a warming climate.** Estimated changes in GPP caused by changes in CF relative to the beginning of the depicted period (1984–1988 mean) (Eq. 11 and "Methods") using CF records from **a** the Climatic Research Unit time-series version 4.08 (CRU TS v4.08) between 1984 and 2023, **b** the international satellite cloud climatology project high-resolution global monthly product (ISCCP-HGM) between 1984 and 2016, **c** the European Centre for Medium-Range Weather Forecasts (ECMWF) fifth generation reanalysis (ERA5) between 1984 and 2024, and **d** the sixth phase Coupled Model Intercomparison Project (CMIP6) 39 models ensemble mean for both historical (1984–2014) and "ssp585" projected (2015–2099) periods. The time series of area-weighted averages for global vegetation areas ("Methods"), arid vegetation areas with humidity index (HI) < 1, and humid vegetation areas with HI > 1 are represented by black, orange, and blue lines, respectively. The annual scale GPP-to-CF sensitivity used for the estimations is derived from Fig. 3a. The shadings represent the 99% confidence interval for the means.

## Methods

### Eddy covariance measurements

We use the FLUXNET2015 dataset[44], which comprises eddy covariance sites worldwide, to quantify in situ GPP and environmental parameters. This dataset provides standardized flux measurements of carbon, water, and energy across 212 sites with rigorous quality assurance, quality control, and gap-filling procedures. To ensure robust statistical analysis in vegetation areas with sufficient data samples, we exclude sites with a data record shorter than 3 years and an average GPP below 0.1 g C m$^{-2}$ d$^{-1}$ over the entire measurement period. As a result, a total of 114 sites are selected for analysis (Supplementary Fig. 23a), with details listed in Supplementary Table 1. Apart from GPP data, PA and SWin are used as environmental variables for mechanistic analysis. Here, in the absence of PAR measurements for FLUXNET data, we use SWin instead because the two variables have an approximate relationship[9,76], i.e., PAR ≈ 0.5 × SWin. Furthermore, additional variables, including air temperature, wind speed, latent heat flux, sensible heat flux, and VPD, are used to calculate PET via the Penman–Monteith equation[77]. This enables the assessment of hydroclimatic conditions in combination with PA.

To investigate the sensitivity of GPP to CF, we separately match sites with the spatially closest ERA5 CF data at 0.25° × 0.25° resolution (daily temporal resolution)[47] and MODIS/Terra CF data at 1° × 1° resolution (MOD08_D3, Version 6.1, daily temporal resolution)[46] for CF information above the sites. For MODIS, in addition to matching total CF, we also match the CF for single-layer liquid and ice clouds, CFliquid and CFice, separately. Since the MODIS data starts later (from 2001 for the first full year of MODIS/Terra) than the measurement at some sites, the available periods for these sites are shortened when matched with MODIS CF, resulting in a final set of 113 sites.

### FLUXCOM GPP datasets

In addition to in situ FLUXNET measurements, we conduct global-scale investigations using GPP datasets from FLUXCOM. The FLUXCOM initiative produces an ensemble of machine learning-based global flux products by upscaling local eddy covariance measurements from tower footprints to global scale maps[45]. In this study, FLUXCOM GPP data are derived from three related products: (1) FLUXCOM-RS V006 (8-daily, 0.083° × 0.083° resolution), where fluxes are forced solely by MODIS satellite data, (2) FLUXCOM-RS_METEO_ERA5 (daily, 0.5° × 0.5° resolution), where fluxes are forced by both MODIS satellite data and meteorological parameters from ERA5, and (3) FLUXCOM-X-BASE (daily, 0.25° × 0.25° resolution), which also incorporates MODIS and

Therefore, from a global perspective, using CF instead of the commonly used combination of cloud-mediated PA and PAR to regress GPP offers some advantages. First, satellite retrievals of surface physical quantities are susceptible to clouds, especially when using optical and infrared sensors. For instance, the clouds and the Earth's radiant energy system (CERES) instrument estimates surface PAR by integrating top-of-atmosphere radiation measurements with auxiliary cloud, aerosol, and atmospheric information through radiative transfer modeling, allowing for uncertain retrieval of surface radiative fluxes under all-sky conditions[75]. Second, compared to multiple linear regression, simply using a linear regression with only one regressor mitigates the uncertainties that arise from the accumulation of errors in multivariable retrievals, while avoiding the complexity and potential instability associated with the interaction of multiple variables. It should be noted that the performance analysis is based on FLUXCOM-RS, since the relationship between CF and GPP at the site scale relies on CF data from reanalysis or satellite products with coarse spatial resolution to approximate actual local cloud conditions. Therefore, ground-based direct CF observations at eddy-covariance sites are needed to represent site-level variability more accurately.

ERA5 forcing data but is based on a newly implemented upscaling framework and an expanded set of flux tower measurements[78]. To minimize potential misinterpretations of the sensitivity of GPP to CF caused by input meteorological variables, we consider FLUXCOM-RS as the benchmark in this study. These three datasets, spanning 2001 to 2020, are ultimately unified to an 8-daily temporal scale with a $1° \times 1°$ spatial resolution for analysis. We also convert the FLUXCOM-RS data to a monthly scale to compare the results of GPP sensitivity to CF across different temporal scales. A key point to note is that our whole global-scale study focuses exclusively on vegetation-covered areas, defined by a multi-year (2001–2020) mean FLUXCOM-RS GPP over $0.1\,g\,C\,m^{-2}\,d^{-1}$ within a grid cell.

FLUXCOM GPP data are employed for the annual scale GPP-to-CF sensitivity estimation. FLUXCOM-RS and FLUXCOM-X-BASE incorporate interannually dynamic remote sensing forcing data, whereas the FLUXCOM-RS_METEO_ERA5 product uses only the mean seasonal cycle of remote sensing data (without interannual dynamics)[78]. Therefore, we only focus on the FLUXCOM-RS and FLUXCOM-X-BASE products for the annual scale analysis, processing them into annual scale $1° \times 1°$ GPP data.

## Remote sensing vegetation indices

The results based on FLUXNET and FLUXCOM GPP data are further evaluated against gridded vegetation indices from satellite remote sensing. Specifically, we employ three other independent remote sensing indicators to estimate vegetation photosynthesis, including GPP, SIF, and VOD.

For remote sensing-based GPP estimation, we use the 8-daily composite data from the MODIS instrument aboard the Terra satellite (MOD17A2HGF, Version 6.1) at a spatial resolution of $0.5° \times 0.5°$, covering the period from 2001 to 2021[79]. To facilitate analysis, the GPP data are averaged to a $1° \times 1°$ resolution. Moreover, we also compute the annual average of the MODIS GPP data for the annual scale analysis (2001–2020). MODIS GPP is derived using a light use efficiency (LUE) model, which assumes that gross photosynthesis is proportional to the amount of absorbed PAR by vegetation[80,81]. A key source of uncertainty arises from the challenge of accurately defining the maximum LUE, particularly in complex and heterogeneous ecosystems[82]. Despite the limitations, the MODIS GPP product remains the most widely used global remote sensing dataset for GPP estimation, owing to its consistent spatial and temporal coverage[83,84].

SIF, emitted by chlorophyll molecules in green plants upon excitation by absorbed sunlight, serves as a direct indicator of photosynthesis[85]. The GOSIF gridded product is derived from discrete SIF retrievals from observations made by the second orbiting carbon observatory (OCO-2) satellite using a machine learning method, combining the enhanced vegetation index (EVI) from MODIS and meteorological reanalysis data from the modern-era retrospective analysis for research and applications, version 2 (MERRA-2)[86]. These estimations from GOSIF have been identified as a reliable predictor of GPP measured from eddy covariance towers[87], even if their predictive capacity comes more from a proper estimation of the absorbed PAR (APAR) rather than the SIF itself. In this study, we use the 8-daily data from the GOSIF product at a $0.05° \times 0.05°$ resolution over the period 2001–2021, and average the data to a $1° \times 1°$ grid.

However, optical remote sensing vegetation indices are often influenced by clouds[88], although multi-day temporal compositing helps mitigate these cloud-related impacts[89,90]. Therefore, we additionally employ the semi-daily X-band VOD data at 10.7 GHz from LPDR v3 for the period 2003–2021[91,92]. This dataset is derived from calibrated microwave brightness temperature records collected by the Advanced Microwave Scanning Radiometer for EOS (AMSR-E) aboard the Aqua satellite and the Advanced Microwave Scanning Radiometer 2 (AMSR2) on the JAXA GCOM-W1 satellite. Microwave remote sensing offers opportunities for monitoring vegetation dynamics, as X-band

VOD is particularly sensitive to upper-canopy water content and stomatal regulation, while also being independent of clouds[23,93]. The LPDR v3 X-band VOD dataset excludes periods with strong precipitation to minimize the impact of canopy-intercepted rainfall on VOD retrievals[91]. Moreover, it provides both daytime and nighttime observations, each providing unique insights into vegetation water dynamics. Daytime VOD (with an equator-crossing overpass time of 13:30 p.m. at nadir) is primarily influenced by plant hydraulics, reflecting the imbalance between transpiration and root-zone water supply. When leaf stomata open during the day, plant water loss through transpiration typically exceeds root-soil water absorption, resulting in a water deficit within the plant[94]. In contrast, nighttime VOD (with an equator-crossing overpass time of 1:30 a.m. at nadir) is driven by root-zone soil moisture replenishment, as transpiration ceases with the closing of leaf stomata after sunset, and exhibits a nearly linear relationship with soil water potential[95,96]. To better capture vegetation photosynthesis dynamics, we calculate the VODndr, which serves as an indicator of the vegetation ability to regulate water loss and uptake in response to environmental conditions[23,95]. Since nighttime VOD reflects soil water recharge and root-zone moisture availability, while daytime VOD is influenced by transpiration-driven water loss, a higher VODndr implies greater soil moisture retention and a reduced transpiration demand, often associated with water stress mitigation and more stable stomatal conductance[23,94,97]. Given that stomatal activity directly controls $CO_2$ uptake, VODndr thus provides insight into vegetation photosynthetic performance. Although VODndr is commonly used to characterize vegetation isohydricity[97], here we use it as a complementary indicator to describe photosynthetic activity. Further validation of VODndr against FLUXNET GPP supports this inference (Supplementary Fig. 24f), while we acknowledge the relatively weak correlation compared to other indicators, reflecting the inherent limitations of this proxy. Nevertheless, VODndr provides complementary information derived from an independent, active remote sensing approach, which is not captured by other datasets. By including VODndr alongside other indicators, we aim to enhance the robustness of our findings while being careful not to over-interpret its specific contributions. The original VOD dataset, available at a semi-daily temporal and $0.25° \times 0.25°$ spatial resolution, is ultimately processed into an 8-daily $1° \times 1°$ VODndr to align with the other datasets.

## Dynamic global vegetation models

We also use monthly GPP estimates from 20 dynamic global vegetation models included in the TRENDYv12 project to assess the sensitivity of GPP to CF from a perspective that excludes climate-vegetation feedbacks[48]. Details of these models are summarized in Supplementary Table 2. Our analysis relies on outputs from simulation scenario S2, which captures temporal variations in climate and $CO_2$ concentration while maintaining a stable preindustrial land cover to eliminate confounding effects from land use change. The simulations are driven by a 6-hourly $0.5° \times 0.5°$ CRU Japanese Reanalysis (JRA) historical climate forcing dataset. To ensure consistency, model outputs (2001–2020)—originally at varying spatial resolutions—are bilinearly interpolated to a common $1° \times 1°$ grid, and the ensemble mean of the 20 models is employed for investigation.

## Satellite cloud retrievals

For the sensitivity analysis of vegetation to clouds, we use daily cloud parameters at a $1° \times 1°$ resolution, derived from measurements by the MODIS instrument aboard the Terra satellite (MOD08_D3, Version 6.1)[46]. Specifically, we consider parameters including CF, CFliquid, CFice, and COT. These parameters capture variations in cloud phase, horizontal cover, and vertical optical thickness, enabling a detailed assessment of their potential effects on vegetation. CFliquid and CFice are the cloud fraction of a single cloud layer only. $\overline{COT}_{cloudy}$ represents the average optical thickness within cloudy areas, while multiplying it

by CF yields the grid-mean COT ($\overline{COT}$), accounting for both cloudy and cloud-free regions (i.e., $\overline{COT} = CF \times \overline{COT}_{cloudy}$). MODIS cloud data from 2001 to 2023 are extracted for different periods and processed into 8-daily, monthly, or annual averages to facilitate sensitivity analysis across different time periods and temporal scales.

To distinguish the responses of vegetation to different cloud types, we use 8-daily joint histograms of COT and CTP from MODIS/Terra (MOD08_E3, Version 6.1)[62], classified according to the ISCCP scheme at a 1° × 1° resolution from 2001 to 2020. The original joint histograms define seven CTP intervals and eight COT intervals. These are further merged into three intervals each, resulting in nine reclassified cloud categories. The categories include: cumulus (COT between 0 and 3.6, CTP between 1100 and 680 hPa), stratocumulus (COT between 3.6 and 23, CTP between 1100 and 680 hPa), stratus (COT between 23 and 150, CTP between 1100 and 680 hPa), altocumulus (COT between 0 and 3.6, CTP between 680 and 440 hPa), altostratus (COT between 3.6 and 23, CTP between 680 and 440 hPa), nimbostratus (COT between 23 and 150, CTP between 680 and 440 hPa), cirrus (COT between 0 and 3.6, CTP between 440 and 0 hPa), cirrostratus (COT between 3.6 and 23, CTP between 440 and 0 hPa), and deep convection (COT between 23 and 150, CTP between 440 and 0 hPa).

### Satellite retrievals of surface solar radiation and precipitation

Satellite observations of surface downwelling solar radiation and precipitation are used for the mechanistic analysis. Surface downwelling solar radiation is derived from the CERES dataset edition 4.1 (SYN1deg Level 3), based in addition to CERES on MODIS Aqua and Terra cloud and aerosol retrievals, with a daily timescale and a spatial resolution of 1° × 1°[98]. Total PAR is obtained by combining direct and diffuse PAR fluxes. The PA dataset used in this analysis is obtained from the Global Precipitation Measurement (GPM) Integrated Multi-satellitE Retrievals for GPM (IMERG) Version 07, with a spatial resolution of 0.1° × 0.1° and a temporal resolution of 30 minutes[99]. Both datasets, covering the period 2001–2020, are processed by averaging to an 8-daily timescale, with IMERG data further averaged to a 1° × 1° spatial resolution.

### Climate data records of CF

To evaluate historical changes in CF, we derive annual average CF from three independent long-term climate data records, each based on ground-based observations, satellite retrievals, and reanalysis, respectively. These include (1) CRU TS v4.08, (2) ISCCP-HGM, and (3) ERA5.

The CRU TS v4.08 dataset integrates observations from comprehensive ground station networks through spatial interpolation, providing monthly records at 0.5° × 0.5° resolution[100]. The ISCCP-HGM combines observations from polar-orbiting and geostationary satellites to generate global monthly cloud parameters at a 1° × 1° spatial resolution[101]. The ERA5 reanalysis offers monthly average data using 4D-Var data assimilation and model forecasts of the ECMWF Integrated Forecast System, with a spatial resolution of 0.25° × 0.25°[47]. The time periods for the annual average CF, 1984–2023 (CRU TS v4.08), 1984–2016 (ISCCP-HGM), and 1984–2024 (ERA5), are individually used for time series analysis. To ensure temporal consistency across all datasets, we extract the annual average CF time series spanning 1984–2014 for the linear trend analysis. Spatial harmonization is achieved through bilinear interpolation to a unified 1° × 1° grid.

### Historical and projected CF simulations from climate models

We use CF simulations from 39 models included in CMIP6[102], with details summarized in Supplementary Table 3, to examine long-term CF changes and trends. The analysis is based on the "historical" experiment driven by all forcings for the period from 1984 to 2014, and the strong-warming future scenario "ssp585" (i.e., Shared Socio-economic Pathway 5 and 2100 climate forcing level of 8.5 W m$^{-2}$)

experiment for the projected period (2015–2099). This scenario provides insights into potential CF changes under a strongly warming climate. All CMIP6 data are obtained at the monthly time scale and bilinearly interpolated to a common 2° × 2° resolution. The assessment of long-term CF changes is conducted using the annual average of the ensemble mean from the 39 models.

### MODIS vegetation types

The Terra and Aqua combined MODIS land cover product (MCD12C1, Version 6.1) with the International Geosphere-Biosphere Programme (IGBP) classification scheme at a 0.05° × 0.05° spatial resolution is employed to define the global vegetation types[103]. The IGBP scheme originally includes 17 land cover types; however, we excluded non-vegetated categories and reclassified the remaining types into nine groups: (1) evergreen needleleaf forest, (2) evergreen broadleaf forest, (3) deciduous needleleaf forest, (4) deciduous broadleaf forest, (5) mixed forest, (6) shrublands (combining closed shrublands and open shrublands), (7) savannas (combining woody savannas and savannas), (8) grasslands, and (9) croplands. Annual data from 2001 to 2020 are averaged and aggregated to a 1° × 1° resolution. Pixels where a specific vegetation type fraction constitutes more than 50% are assigned that type. The map of the vegetation types is shown in Supplementary Fig. 25.

### Definition of HI

We define the HI[9], in some references also known as aridity index[104,105], to describe the hydroclimatic conditions. HI is calculated as the ratio of long-term mean annual PA to PET, based on monthly PA and PET data from the CRU TS v4.08 (2001–2020) at a 0.5° × 0.5° spatial resolution[100]. Therefore, lower HI values indicate more arid conditions, while higher values correspond to more humid climates. Finally, HI is averaged to a 1° × 1° resolution. Supplementary Fig. 23b shows the map of HI. In addition, for each FLUXNET site, we calculate the HI as the ratio of mean PA to PET over the entire measurement period (Supplementary Fig. 23a).

### Meteorological active season

The meteorological active season, also referred to as the growing season[106,107], is defined as the period encompassing months with a long-term (2001–2020) average T2m > 0 °C[108]. In arid and semi-arid ecosystems (HI < 1), this period is further constrained to months where cumulative PA lies between 10 and 90% of the annual total PA. Incorporating a PA threshold in the definition of the meteorological active season for water-limited environments accounts for potential inactive vegetation phases, even when T2m > 0 °C, due to water deficits[109]. For arid and semi-arid grids in the Southern Hemisphere, PA accumulation begins in July and ends in June of the following year. T2m and PA are obtained from the CRU TS v4.08 datasets[100], bilinearly interpolated to a 1° × 1° resolution.

### Vegetation sensitivity to clouds

We initially preprocess the data by removing long-term trends and seasonal cycles through a two-step approach: linear detrending followed by Z-score standardization. For example, with the site-scale FLUXNET data, we assume an interannual linear trend for the same calendar day across multiple years, modeled as:

$$X_t = a \times t + b + \varepsilon_t \qquad (1)$$

where $X_t$ represents the data for a specific calendar day in year $t$, $a$ and $b$ are the estimated regression coefficients, and $\varepsilon_t$ denotes the residuals.

We first remove interannual trends by fitting a linear regression model to the data for the same calendar day across multiple years. The

detrended value $X'_t$ is then given by:

$$X'_t = X_t - (a \times t + b) \quad (2)$$

After detrending, we compute the Z-score for the same calendar day across multiple years to further eliminate seasonal effects:

$$X^*_t = \frac{X'_t - \mu}{\sigma} \quad (3)$$

where $\mu$ and $\sigma$ represent the mean and standard deviation of $X'$ for the same calendar day across years. The final standardized value, $X^*$, ensures comparability across all data by removing both interannual and seasonal variations.

Then, we apply a univariate linear regression to estimate the sensitivity of vegetation dynamics to clouds using the standardized values. This approach simplifies the analysis by focusing on a single factor, avoiding multicollinearity issues, and providing a clearer, more interpretable measure of the relationship between vegetation and clouds. The sensitivity of vegetation indices to clouds is estimated as the following equation:

$$S = \frac{\partial \text{Veg}^*}{\partial \text{Cld}^*} \quad (4)$$

where $\text{Veg}^*$ and $\text{Cld}^*$ are, respectively, the standardized vegetation indices (e.g., GPP, SIF, and VODndr) and the standardized cloud properties (e.g., CF, CFliquid, CFice, and $\overline{\text{COT}}$) as derived from Eq. 3. $S$ represents the sensitivity of vegetation indices to clouds for the standardized data, obtained by performing a linear regression between the time series of $\text{Veg}^*$ and $\text{Cld}^*$. In addition, the P-value for the sensitivity is calculated using a Student's t test, which helps assess the statistical significance of the estimated sensitivity.

A similar approach is also applied to global-scale data analysis using 8-daily or monthly averages. For FLUXNET data, only sites with records spanning more than 10 years are detrended in the first step. Additional sensitivity tests, including cases where all sites are detrended and where no sites undergo detrending, are presented in Supplementary Fig. 26. These tests indicate that detrending has only a minor influence on the GPP*-to-CF* sensitivity analysis for FLUXNET data.

To assess the sensitivity of GPP to the CF of each cloud type, we perform a multiple linear regression, using the CF* of all cloud types as the regressors and GPP* as the response variable, as described by the following equation:

$$\text{GPP}^* = \beta_0 + \sum_{i=1}^{n} (\beta_{\text{CF}_i} \times \text{CF}_i^*) + \varepsilon \quad (5)$$

where $\text{CF}_i$ represents the CF for cloud type $i$, with $i = 1, 2, 3 \ldots, n$. Based on our reclassification of MODIS/Terra joint histograms of COT and CTP data, here $n = 9$. $\beta_0$ is the intercept, and $\beta_{\text{CF}_i}$ represent the partial sensitivity of GPP* to $\text{CF}_i^*$. $\varepsilon$ denotes the residual. The analysis is conducted using 8-daily data for all variables. We compute the VIF for each regressor and consider the multi-collinearity negligible if the maximum VIF is less than 5. Since all the data are standardized, the values of the partial sensitivities can be directly compared. Hence, we calculate the relative contribution of cloud type $i$ ($\text{RC}_i$) to the GPP*-to-CF* sensitivity using the following equation:

$$\text{RC}_i = \frac{\beta_{\text{CF}_i}}{\sum_{i=1}^{n} \beta_{\text{CF}_i}} \quad (6)$$

Beyond analysis based on standardized values (i.e., GPP*-to-CF* sensitivity), the global pattern of GPP-to-CF sensitivity is also estimated on an annual scale. This approach, based on raw value estimates, provides a basis for predicting future cloud-driven GPP changes. Following the method of refs. 110,111, this sensitivity is quantified as the partial derivative derived from a multiple linear regression model. The model captures year-to-year variations in GPP related to year-to-year variations in CF while accounting for year-to-year variations in potentially confounding climate factors, for which we selected T2m and $q$, within the attribution framework:

$$\delta\text{GPP} = \beta_0 + \beta_{\text{CF}} \times \delta\text{CF} + \beta_{\text{T2m}} \times \delta\text{T2m} + \beta_q \times \delta q + \varepsilon \quad (7)$$

where the $\delta$ operator represents the difference between two consecutive years (year-to-year variation). $\beta_0$ is the intercept, and the other $\beta$ values denote the partial sensitivities of $\delta\text{GPP}$ to the respective variables. $\varepsilon$ represents the residual. The regression is based on annual scale data from 2001 to 2020. CF data are sourced from MODIS/Terra, while GPP data are obtained from FLUXCOM-RS, FLUXCOM-X-BASE, and MODIS/Terra, providing estimates based on multiple GPP datasets. The annual average $1° \times 1°$ meteorological factors of T2m and $q$ are derived from CRU JRA v2.5 6-hourly $0.5° \times 0.5°$ meteorological reanalysis[112]. This year-to-year variation approach helps to disentangle the resulting signal from potential long-term dependencies on covariates, such as the combined effect of rising temperatures and increasing $CO_2$ concentrations on long-term GPP and CF trends[43,111]. Moreover, accounting for background climate factors constrains the collaborative influence of environmental factors on both clouds and vegetation[29]. Considering potential multi-collinearity among regressors involved in the multiple linear regression, for instance, CF may be correlated with T2m and $q$, we compute the VIF for each regressor[29]. If the maximum VIF of all regressors is less than 5, we deem multi-collinearity negligible. $R^2$ of the regressors in Eq. 7 and the incremental variance explained of each regressor are computed. We quantify the incremental variance explained of each predictor as the change in the $R^2$ within a stepwise regression framework. Specifically, a baseline model comprising all other predictors is first established, followed by the individual addition of the target variable. The incremental variance explained is then calculated as the increase in $R^2$ attributable to the inclusion of that variable.

To enhance the reliability and spatial consistency of the sensitivity estimates, we apply a spatial moving window to perform the regression analysis for the centered grid cells[29,111]. Specifically, for each pixel-level analysis, we spatially aggregate annual values from the surrounding $3° \times 3°$ grids, effectively increasing the sample size by a factor of nine. This step is crucial, as the use of annual scale variables results in a limited sample size for conducting partial regression analysis using Eq. 7. To evaluate the impact of spatial aggregation, we perform three additional tests (Supplementary Fig. 27). First, we repeat the analysis on individual pixels without spatial aggregation to isolate the effects of temporal variability. Second, we apply the spatial aggregation method using $5° \times 5°$ and $7° \times 7°$ moving windows, respectively. The sensitivities obtained from these different spatial aggregation approaches exhibit similar spatial patterns and magnitudes. Thus, we conclude that the results are minimally sensitive to the choice of spatial aggregation method and ultimately adopt the $3° \times 3°$ moving window as the primary approach in our main analysis.

### Time-lagged sensitivity

Since regression-based relationships do not establish causality, in addition to validation using offline simulations from TRENDY, we complement the analysis with a data-driven approach to reveal the causal relationship between GPP and CF. This is explored through a time-lagged sensitivity using 8-daily data, expressed as:

$$S(\tau) = \frac{\partial \text{GPP}^*(t)}{\partial \text{CF}^*(t+\tau)} \quad (8)$$

where $t$ denotes the time series for 8-day intervals, ranging from 1 to $T$, where $T$ is the total number of 8-day periods. $GPP^*(t)$ and $CF^*(t+\tau)$ are time series of the same length, representing the standardized GPP and CF, respectively, over an 8-daily temporal resolution. The time lag $\tau$ is expressed in units of 8-day intervals. The time-lagged $GPP^*(t) - to - CF^*(t+\tau)$ sensitivity $S(\tau)$, which quantifies the sensitivity after applying the time lag $\tau$, is obtained by performing a linear regression between the time series of $GPP^*(t)$ and $CF^*(t+\tau)$. To ensure valid computations, the analysis considers different index ranges based on $\tau$: for $\tau > 0$, the valid range for $t$ is $1 \le t \le T - \tau$, whereas for $\tau < 0$, the range is $1 - \tau \le t \le T$, ensuring that all required values remain within the dataset. This formulation allows us to investigate how shifts in CF over time are related to the GPP dynamics, providing insights into the causal relationship between these two variables. By examining the time-lagged sensitivity across different values of $\tau$ (e.g., −2, −1, 1, and 2), we can assess the temporal responsiveness of GPP to CF variations. This approach enhances our understanding of the time-dependent nature of GPP and CF interaction, which is critical for identifying causal links beyond transient correlations[113].

## Vegetation sensitivity to precipitation and radiation

Following the same approach in Eq. 4, we separately estimate the $GPP^*$-to-PA$^*$ sensitivity ($\frac{\partial GPP^*}{\partial PA^*}$) and the $GPP^*$-to-PAR$^*$ sensitivity ($\frac{\partial GPP^*}{\partial PAR^*}$) to assess vegetation sensitivity to cloud-related precipitation and radiation. Additionally, we investigate GPP sensitivities to precipitation and radiation using a multiple linear regression, with PA$^*$ and PAR$^*$ as the regressors, as described by the following equation:

$$GPP^* = \beta_0 + \beta_{PA} \times PA^* + \beta_{PAR} \times PAR^* + \varepsilon \qquad (9)$$

where $\beta_0$ is the intercept, and $\beta_{PA}$ and $\beta_{PAR}$ represent the partial sensitivity of $GPP^*$ to PA$^*$ and $GPP^*$ to PAR$^*$, respectively. $\varepsilon$ denotes the residual. The analysis is conducted using 8 daily data for all variables. The difference between $\beta_{PA}$ and $\beta_{PAR}$ ($\beta_{PA} - \beta_{PAR}$) provides insight into the relative importance of precipitation and radiation in influencing GPP. Since the model includes only two regressors, the VIF values for PA$^*$ and PAR$^*$ are identical. We consider the multi-collinearity negligible if the VIF is less than 5. The study employs 8 daily gridded data for analysis. In the case of FLUXNET, the analysis is based on daily measurements, with SWin used as a substitute for PAR.

In addition, based on 8 daily data, we construct a multiple linear regression model using PA* at various lag times as predictors to quantify the delayed effect of GPP* in response to PA*. The model is formulated as:

$$GPP^*(t) = \beta_0 + \sum_{\tau=0}^{l} \beta_\tau \times PA^*(t+\tau) + \varepsilon \qquad (10)$$

where $t$ denotes the time series for 8-day intervals, ranging from 1 to $T$, where $T$ is the total number of 8-day periods. All variables are time series of the same length. It should be noted that the unit of lag time $\tau$ is an 8-day interval. The longest lag $l$ is set to −8, corresponding to a maximum lag time of 64 days, which means that we use PA$^*(t+\tau)$ at lag times $\tau = 0, -1, \ldots, -8$ in the regression. We also test the longest lag $l = -9$ and $l = -10$. Both cases suggest that nearly 99% of pixels globally exhibit delayed effects shorter than two months ($\tau \ge -8$). Therefore, the use of $l = -8$ is appropriate, as it captures the substantial lagged effects while excluding negligible contributions. $\beta_0$ is the intercept, and $\varepsilon$ denotes the residual. The lag time $\tau$ of the maximum partial regression coefficient $\beta_\tau$ with statistical significance ($P < 0.001$) is identified as the effective response period of GPP* to PA*.

## Effects of long-term CF changes and trends on GPP

To empirically estimate long-term CF-driven changes in GPP, the variations or linear trends in CF are multiplied by the previously derived

$\beta_{CF}$ in Eq. 7. However, it should be noted that these long-term trend estimates carry uncertainties, as they assume that GPP sensitivity to CF remains constant over time[29]. Nevertheless, the focus of this study is on the long-term shifts in the spatial patterns of CF-driven GPP, with changes in the values of $\beta_{CF}$ under climate change expected to introduce only minor biases in the spatial patterns. We incorporate multiple long-term cloud climate records for the estimates, applying the following equations:

$$\Delta GPP^{CF} = \beta_{CF} \times \Delta CF \qquad (11)$$

$$\gamma GPP^{CF} = \beta_{CF} \times \gamma CF \qquad (12)$$

where $\Delta CF$ indicates CF changes relative to the mean of the initial five years, $\Delta GPP^{CF}$ denotes the corresponding changes in GPP attributed to CF alterations, $\gamma CF$ refers to the linear trend in long-term CF, and $\gamma GPP^{CF}$ is the estimated trend in CF-driven GPP. The $\beta_{CF}$ is bilinearly interpolated to a $2° \times 2°$ grid when applying to the analysis with CF data from CMIP6. When calculating the linear trend of long-term CF, we also compute the $P$-value using a Student's $t$ test to assess its statistical significance.

## Data availability

The FLUXNET2015 observations can be obtained from https://fluxnet.org/data/fluxnet2015-dataset/. The FLUXCOM-RS 8-daily GPP datasets and FLUXCOM-RS_METEO_ERA5 daily GPP datasets are available on request to Martin Jung (mjung@bgc-jena.mpg.de). The FLUXCOM-X-BASE daily GPP datasets can be obtained from https://meta.icos-cp.eu/collections/AYj7-lwcdCLnBXJDoscxQZou. The MODIS/Terra MOD17-A2HGF Version 6.1 8-daily GPP data can be obtained from https://lpdaac.usgs.gov/products/mod17a2hgfv061/. The MODIS/Terra MOD08_D3 Version 6.1 daily cloud parameters can be obtained from https://ladsweb.modaps.eosdis.nasa.gov/archive/allData/61/MOD08_D3/. The MODIS/Terra MOD08_E3 Version 6.1 8-daily cloud parameters can be obtained from https://ladsweb.modaps.eosdis.nasa.gov/archive/allData/61/MOD08_E3/. The MODIS land cover product (MCD12C1) Version 6.1 can be obtained from https://lpdaac.usgs.gov/products/mcd12c1v061/. The GOSIF 8-daily SIF data can be accessed at https://climatedataguide.ucar.edu/climate-data/global-dataset-solar-induced-chlorophyll-fluorescence-gosif. The LPDR v3 semi-daily VOD data can be downloaded at http://files.ntsg.umt.edu/data/LPDR_v3/GeoTif/. The monthly GPP simulations from TRENDYv12 models are available at https://mdosullivan.github.io/GCB/. The CERES SYN1deg Level 3 daily radiation can be downloaded from https://ceres.larc.nasa.gov/data/. The IMERG 30-minutes precipitation dataset can be accessed at https://disc.gsfc.nasa.gov/datasets/GPM_3IMERGHH_07/summary?keywords=IMERG. The CMIP6 monthly outputs of CF are available at https://esgf-data.dkrz.de/search/cmip6-dkrz/. The CRU TS v4.08 monthly CF data can be obtained from https://catalogue.ceda.ac.uk/uuid/715abce1604a42f396f81db83aeb2a4b/. The CRU JRA v2.5 6-hourly meteorological reanalysis can be accessed at https://catalogue.ceda.ac.uk/uuid/43ce517d74624a5ebf6eec5330cd18d5/. The ISCCP-HGM monthly CF data are available at https://www.ncei.noaa.gov/products/climate-data-records/cloud-properties-isccp. The ERA5 monthly average reanalysis of CF can be accessed at https://cds.climate.copernicus.eu/datasets/reanalysis-era5-single-levels-monthly-means?tab=download. The ERA5 post-processed daily reanalysis of CF can be accessed at https://cds.climate.copernicus.eu/datasets/derived-era5-single-levels-daily-statistics?tab=download.

## Code availability

The codes associated with the main figures in this study are available at https://zenodo.org/records/17601937 (ref. 114).

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

## Acknowledgements

G.D. and M.R. acknowledge funding by the European Research Council (ERC) Synergy Grant "understanding and modeling the Earth system with machine learning (USMILE)" under the European Union's Horizon 2020 research and innovation program (grant agreement no. 855187).

## Author contributions

J.Q. and H.L. designed the research. H.L. performed the research and drafted the paper. H.L., A.B., M.R., G.D., J.K., and J.Q. contributed to the analysis and interpretation of the results, as well as revising the paper.

## Funding

## Competing interests

The authors declare no competing interests.

## Additional information

**Supplementary information** The online version contains Supplementary material available at https://doi.org/10.1038/s41467-026-69480-3.

