## [Transparent Peer Review File · Nature Communications]

Hydroclimate shapes photosynthetic sensitivity to cloud cover across global terrestrial ecosystems

Corresponding Author: Dr Hao Luo

Version 0:

Reviewer comments:

Reviewer #2

(Remarks to the Author)

GENERAL COMMENTS

Luo and colleagues present research that disentangles the influence of cloud cover on gross primary productivity. Using multiple remote-sensed, modeled, and observational datasets, they demonstrate a systematic pattern: in water-limited environments, cloud cover increases GPP through increased rainfall, while in energy-limited environments, cloud cover decreases GPP by reducing photosynthetically active radiation. The use of offline TRENDY models to assess causality is particularly clever. I commend the authors on their robust approach, thoroughly testing the sensitivity of their results to different data products. My comments below aim to improve readability and encourage consideration of implications not currently addressed in the manuscript.

Implications for climate projections

Hydroclimate changes: How do these findings change in the context of future hydrologic cycle changes? Held & Soden (2006) demonstrates that while the atmosphere's water-holding capacity increases by about 7% per degree of warming, global precipitation increases more slowly—by roughly 2–4% per degree—due to energetic constraints on the hydrological cycle. Could this influence the results in Figure 4d? How do patterns in future hydroclimate changes affect your interpretation of Figure 4d? I'm not suggesting additional analyses, but rather some consideration that future changes to cloud cover are inextricably linked to hydroclimate.

Cloud representation in models: Clouds are among the most challenging components to model in the climate system due to their small-scale processes, often leading to unrealistic parameterizations. This warrants discussion when addressing projected changes.

Readability

This dense manuscript contains extensive detail that sometimes obscures the core message, particularly in the Discussion. I suggest:

1. Moving "Examining different vegetation types" and "Comparison with independent datasets" to supplementary material under a new section titled "Sensitivity Analyses"
2. Renaming the "Results" section to "Results and Discussion" to better reflect its content

These changes would streamline the text, helping readers follow the main narrative without getting lost in excessive detail.

Additionally, Figure 4D's aspect ratio needs adjustment—it appears stretched compared to Figures 4a-c.

LINE BY LINE COMMENTS

L24-27 & L390-394: The term "spatial shift" is confusing and requires multiple readings for clarity. Given this conclusion's importance, consider simpler wording such as: "GPP is expected to decline in arid regions and increase in humid regions"

due to cloud fraction changes." This key takeaway deserves maximum clarity.

REFERENCES

Held, Isaac M., and Brian J. Soden. "Robust responses of the hydrological cycle to global warming." *Journal of climate* 19.21 (2006): 5686-5699.

Reviewer #3

(Remarks to the Author)

Water is a limiting factor for vegetation growth in arid regions, while energy is the constraint in humid regions. The authors further employed cloud fraction as a physical variable to dissect the influence of two independent climatic factors—energy and water—on vegetation photosynthesis. The paper comprehensively utilizes multi-source observational data and model simulations, featuring rigorous analytical methods, a complete chain of reasoning, and thorough validation. It provides new insights into the relationships among energy, water, and vegetation. On this basis, the study predicts that under future climate change, reduced cloud cover may lead to a shift in gross primary productivity (GPP) from arid to humid regions. This finding offers an important warning regarding future changes in the spatial pattern of carbon sinks and ecosystem functions. However, I have several major concerns which may require thoroughly editing before publication is granted.

First, the discussion on cloud fraction as a proxy variable requires strengthening—specifically, the extent to which cloud fraction can reliably represent precipitation. Although the paper provides strong indirect evidence through time-lag analysis, it is recommended to add a dedicated paragraph in the discussion to directly address this issue. In addition, the analysis of cloud types (liquid vs. ice, COT, ISCCP classes) adds depth. However, it somewhat reinforces a question the reader might have: if liquid clouds and high COT are the primary drivers, why is total CF the best variable? The authors argue that total CF is sufficient, which is likely true for a first-order global analysis, but the paper would be significantly strengthened by a direct comparison.

Secondly, FLUXCOM-X-BASE shows a positive sensitivity of GPP to cloud fraction in evergreen broadleaf forests, which contradicts the results from FLUXCOM-RS. The authors attribute this discrepancy to uncertainties in the representation of water processes within the FLUXCOM framework. This explanation remains somewhat vague; a more in-depth exploration of the possible causes is recommended. For example, does the use of meteorological drivers in X-BASE introduce additional collinearity? Or does the structure of its machine learning model respond differently to radiation—particularly diffuse light—compared to the remote sensing (RS) product?

Thirdly, the argument that CF is an effective proxy for the combination of PAR and PA is robust on a global, climatic scale. However, the manuscript could more openly acknowledge a key limitation: at the instantaneous or daily scale, the relationship between CF and precipitation is weak and non-linear. Similarly, the conclusion about a spatial shift in GPP is based on a robust multi-dataset consensus on declining CF. However, the projection figure (Fig. 4) relies on the annual-scale sensitivity (β CF) derived from the recent past (2001-2020) remaining constant throughout the 21st century. This is a strong assumption. The authors should explicitly state this as a key uncertainty in the discussion. Could β CF itself change with increasing CO₂, VPD, or vegetation acclimation? A brief discussion on this would improve the paper's balance.

Finally, given that the Earth has undergone global dimming and brightening—a phenomenon closely linked to changes in cloud cover and aerosols—how consistent are the paper's results with these large-scale trends? It is advised to include relevant discussion on this point.

A few specific comments are given below.

The numbers in front of the comments indicate line number.

1. L121-125. The paragraph describing the TRENDY models could be clearer. It correctly states that these models do not use cloud as a direct forcing factor, yet the analysis correlates their GPP output with MODIS cloud fraction (CF). While this is a valid methodological approach, the text should be rephrased to avoid any potential misunderstanding that cloud data is used as an input to the models themselves.

2. L194. The term 'grid mean COT' is ambiguous. It should be explicitly defined—for example, as "grid-mean cloud optical thickness" (τ), where $\tau = CF \times COT_{avg}$, and COT_{avg} represents the average cloud optical thickness over cloudy pixels only.

3. L495. The use of VOD_{ndr}, which has relatively weak correlation with GPP (as shown in Suppl. Fig. 14f), calls for a more cautious interpretation of the resulting findings.

4. L1064. Legend in panel a is unclear. Do the values 0.1, -0.1, and -0.3 correspond to all sites collectively? It is also difficult to distinguish between the representations of -0.1 and -0.3. Clarification is needed.

Version 1:

Reviewer comments:

Reviewer #2

(Remarks to the Author)

The authors have engaged substantively with every comment, and the revised manuscript reflects their careful attention to detail. I believe it would make an excellent contribution to *Nature Communications*.

Reviewer #3

(Remarks to the Author)

The authors have addressed previous critiques with substantial revisions, and I recommend publication after the following important corrections are made.

1. Title. The manuscript focuses exclusively on terrestrial vegetated areas and excludes oceanic ecosystems involving photosynthesis. Therefore, the term "global ecosystems" is misleading and should be clarified, for example, as "global terrestrial ecosystems."
2. Abstract. It is unclear how long the datasets cover that support the finding that "the sensitivity of photosynthesis to cloud cover is spatially shaped by the hydroclimate." Additionally, in the conclusion, I recommend clearly stating that the further examination is based on "CMIP6 projections under the SSP585 scenario (2015–2099)." Please also change "GPP is expected to decline" to "GPP is projected to decline."
3. Fig. 4. The changes are presented relative to the period 1984–1988. Could the authors explain why this specific period was chosen instead of the longer "historical" experiment period (e.g., 1984–2014)?
4. Line 118. Please replace "S2" with a clearly defined term, such as "the high-emission pathway (SSP5-8.5)."
5. Line 769. The longest lag l is set to -8 . How was the maximum lag time of 64 days determined? Please clarify the rationale for this choice.

Responses to comments on “Hydroclimate shapes photosynthetic sensitivity to cloud cover across global ecosystems” (NCOMMS-25-69717-T)

Hao Luo^{1*}, Ana Bastos², Markus Reichstein³, Gregory Duveiller³,
Jan Kretzschmar¹, Johannes Quaas^{1,4}

¹Leipzig Institute for Meteorology, Leipzig University, 04103, Leipzig, Germany

²Institute for Earth System Science and Remote Sensing, Leipzig University, 04103, Leipzig, Germany

³Max Planck Institute for Biogeochemistry, 07745, Jena, Germany

⁴German Centre for Integrative Biodiversity Research (iDiv) Halle-Jena-Leipzig, 04103, Leipzig, Germany

*Corresponding author. Email: hao.luo@uni-leipzig.de

We are grateful to the two reviewers for their positive, thoughtful, and constructive comments, which are very helpful in improving the quality of this manuscript. We have thoroughly revised the manuscript to address all comments with point-by-point responses. We hope our revised version will receive favourable consideration and meet the journal’s requirements. The reviewers’ comments are reproduced (*black, italic*) along with our replies (*blue*) and changes made to the text (*red*). All the authors have read the revised manuscript and agreed with the submission in its current form.

Responses to Reviewer #1

Reviewer #1 (Remarks to the Author):

GENERAL COMMENTS

Luo and colleagues present research that disentangles the influence of cloud cover on gross primary productivity. Using multiple remote-sensed, modeled, and observational datasets, they demonstrate a systematic pattern: in water-limited environments, cloud cover increases GPP through increased rainfall, while in energy-limited environments, cloud cover decreases GPP by reducing photosynthetically active radiation. The use of offline TRENDY models to assess causality is particularly clever. I commend the authors on their robust approach, thoroughly testing the sensitivity of their results to different data products. My comments below aim to improve readability and encourage consideration of implications not currently addressed in the manuscript.

Response:

We would like to thank the reviewer for taking the time to review this manuscript and for providing positive comments and constructive suggestions.

Implications for climate projections

Comment 1.1:

Hydroclimate changes: How do these findings change in the context of future hydrologic cycle changes? Held & Soden (2006) demonstrates that while the atmosphere's water-holding capacity increases by about 7% per degree of warming, global precipitation increases more slowly—by roughly 2–4% per degree—due to energetic constraints on the hydrological cycle. Could this influence the results in Figure 4d? How do patterns in future hydroclimate changes affect your interpretation of Figure 4d? I'm not suggesting additional analyses, but rather some consideration that future changes to cloud cover are inextricably linked to hydroclimate.

Response:

This is a valuable suggestion which we have followed. We have now added discussions about the uncertainties due to the hydroclimate changes as suggested by the reviewer.

Changes in Manuscript:

“Moreover, under a warming climate, future changes in clouds, radiation, precipitation, and vegetation are expected to co-evolve with adjustments in the hydrological cycle. Here, we highlight several potential processes that respond to climate change: (i) the atmosphere's water-holding capacity increases by about

7 % °C⁻¹, while global precipitation rises more slowly by roughly 2–4 % °C⁻¹ due to energetic constraints⁷¹; (ii) cloud vertical structure is projected to change, featuring a faster decline in liquid clouds than in ice clouds over land, thereby influencing cloud radiative effects³³; (iii) GPP responds negatively to increasing VPD while exhibiting a nonlinear response to soil moisture¹⁸; (iv) thermal acclimation buffers photosynthesis against warming, reducing increases in photosynthesis under cold climates and limiting declines in photosynthesis under hot climates⁷². Collectively, these adjustments imply that cloud and vegetation characteristics will vary according to regional hydroclimate, potentially altering the radiative and precipitative impacts on GPP and introducing uncertainties in the estimation of CF-driven GPP changes shown in Fig. 4. Therefore, as our projections assume that the GPP-to-CF sensitivity inferred from the recent past (2001–2020) remains constant throughout the 21st century, future studies should aim to quantify the potential uncertainties under changing CO₂, VPD, hydrological cycle, and vegetation acclimation.” [Lines 291–306 in the “Track Changes” version]

References:

- 18 Fu, Z. *et al.* Critical soil moisture thresholds of plant water stress in terrestrial ecosystems. *Science Advances* **8**, eabq7827, doi:10.1126/sciadv.abq7827 (2022).
- 33 Luo, H., Quaas, J. & Han, Y. Examining cloud vertical structure and radiative effects from satellite retrievals and evaluation of CMIP6 scenarios. *Atmospheric Chemistry and Physics* **23**, 8169–8186, doi:10.5194/acp-23-8169-2023 (2023).
- 71 Held, I. M. & Soden, B. J. Robust responses of the hydrological cycle to global warming. *Journal of Climate* **19**, 5686–5699, doi:10.1175/JCLI3990.1 (2006).
- 72 Schneider, P. D., Gessler, A. & Stocker, B. D. Global photosynthesis acclimates to rising temperatures through predictable changes in photosynthetic capacities, enzyme kinetics, and stomatal sensitivity. *Journal of Advances in Modeling Earth Systems* **17**, e2024MS004789, doi:10.1029/2024MS004789 (2025).

Comment 1.2:

Cloud representation in models: Clouds are among the most challenging components to model in the climate system due to their small-scale processes, often leading to unrealistic parameterizations. This warrants discussion when addressing projected changes.

Response:

We thank the reviewer for this constructive suggestion. We agree with the reviewer that clouds remain one of the most challenging components to represent in climate models, as their microphysical and

convective processes are parameterized and thus subject to uncertainties. We have added a statement to acknowledge that these limitations in cloud representation can influence the projected changes in cloud fraction and the associated vegetation responses.

Changes in Manuscript:

“It should be noted that clouds remain one of the most uncertain components in climate models because their sub-grid processes are not explicitly resolved and rely on imperfect parameterizations^{69,70}. These limitations may contribute to uncertainties in projected CF changes and, consequently, in the estimated vegetation responses.” [Lines 288–291 in the “Track Changes” version]

References:

- 69 Konsta, D. et al. Low-level marine tropical clouds in six CMIP6 models are too few, too bright but also too compact and too homogeneous. *Geophysical Research Letters* **49**, e2021GL097593, doi:10.1029/2021GL097593 (2022).
- 70 Klein, S. A. et al. Are climate model simulations of clouds improving? An evaluation using the ISCCP simulator. *Journal of Geophysical Research: Atmospheres* **118**, 1329-1342, doi:10.1002/jgrd.50141 (2013).

Readability

Comment 1.3:

This dense manuscript contains extensive detail that sometimes obscures the core message, particularly in the Discussion. I suggest:

1. *Moving "Examining different vegetation types" and "Comparison with independent datasets" to supplementary material under a new section titled "Sensitivity Analyses"*
2. *Renaming the "Results" section to "Results and Discussion" to better reflect its content*

These changes would streamline the text, helping readers follow the main narrative without getting lost in excessive detail.

Response:

Excellent help by the reviewer! The overall structure has now been revised as suggested.

Changes in Manuscript:

1. “Results and Discussion” [Line 87 in the “Track Changes” version]
2. “Sensitivity analysis” [Page 2 in the Supplementary Information]

- “Further sensitivity investigations, including examination of different vegetation types and comparison with independent datasets, are discussed in the Supplementary Materials.” [Lines 102–104 in the “Track Changes” version]

Comment 1.4:

Additionally, Figure 4D's aspect ratio needs adjustment—it appears stretched compared to Figures 4a-c.

Response:

Thank you! We have adjusted Figure 4D's aspect ratio as suggested (Fig. R1.1), so that one unit of time (year) has the same length across panels. Other figures with the same issue have also been adjusted (Extended Data Fig. 7, Supplementary Fig. 26 and Fig. 28).

Figure R1.1. Shifts in gross primary productivity (GPP) from arid to humid regions driven by cloud fraction (CF) changes in a warming climate.

LINE BY LINE COMMENTS

Comment 1.5:

L24-27 & L390-394: The term "spatial shift" is confusing and requires multiple readings for clarity. Given this conclusion's importance, consider simpler wording such as: "GPP is expected to decline in arid regions and increase in humid regions due to cloud fraction changes." This key takeaway deserves maximum clarity.

Response:

We thank the reviewer for pointing this out. We have clarified these statements as suggested.

Changes in Manuscript:

“The findings indicate that, under a warming climate, GPP is expected to decline in arid regions and increase in humid regions due to changes in cloud cover, suggesting an exacerbation of regional disparities in ecosystem functions.” [Lines 24–28 in the “Track Changes” version]

“Finally, based on the annual scale spatially resolved sensitivity metric of GPP to CF, we find for a warming climate, GPP is expected to decrease in arid regions and increase in humid regions due to the decline in global average CF.” [Lines 430–432 in the “Track Changes” version]

REFERENCES

Held, Isaac M., and Brian J. Soden. "Robust responses of the hydrological cycle to global warming." Journal of climate 19.21 (2006): 5686-5699.

Responses to Reviewer #2

Reviewer #2 (Remarks to the Author):

Water is a limiting factor for vegetation growth in arid regions, while energy is the constraint in humid regions. The authors further employed cloud fraction as a physical variable to dissect the influence of two independent climatic factors—energy and water—on vegetation photosynthesis. The paper comprehensively utilizes multi-source observational data and model simulations, featuring rigorous analytical methods, a complete chain of reasoning, and thorough validation. It provides new insights into the relationships among energy, water, and vegetation. On this basis, the study predicts that under future climate change, reduced cloud cover may lead to a shift in gross primary productivity (GPP) from arid to humid regions. This finding offers an important warning regarding future changes in the spatial pattern of carbon sinks and ecosystem functions.

However, I have several major concerns which may require thoroughly editing before publication is granted.

Response:

We would like to thank the reviewer for taking the time to review this manuscript and for providing valuable comments and constructive suggestions.

Comment 2.1:

First, the discussion on cloud fraction as a proxy variable requires strengthening—specifically, the extent to which cloud fraction can reliably represent precipitation. Although the paper provides strong indirect evidence through time-lag analysis, it is recommended to add a dedicated paragraph in the discussion to directly address this issue. In addition, the analysis of cloud types (liquid vs. ice, COT, ISCCP classes) adds depth. However, it somewhat reinforces a question the reader might have: if liquid clouds and high COT are the primary drivers, why is total CF the best variable? The authors argue that total CF is sufficient, which is likely true for a first-order global analysis, but the paper would be significantly strengthened by a direct comparison.

Response:

We thank the reviewer for raising this important point. We have now added a direct global assessment of the relationships between cloud fraction (CF) and precipitation amount (PA), as well as between CF and photosynthetically active radiation (PAR). All data are standardized by removing long-term trends and seasonal cycles (marked with an asterisk). The results show statistically strong positive correlations between PA* and CF* and strong negative correlations between PAR* and CF* (Fig. R2.1), thereby directly supporting the use of total CF as an effective combined proxy for precipitation and radiation in this study.

Of course, as the reviewer rightly noted, the use of total CF is appropriate for a first-order global analysis, particularly because it is more consistently and accurately detected by satellites. Although we have attempted examinations of cloud types, these findings lay the groundwork for further investigations into cloud-type-specific influences in a more comprehensive and quantitative manner.

Figure R2.1. Correlations between photosynthetically active radiation (PAR) and cloud fraction (CF), as well as between precipitation amount (PA) and CF. (a) Map of the correlation coefficient between PAR* and CF*. (b) Map of the correlation coefficient between PA* and CF*. The analysis is based on the periods between 2001 and 2020 using 8-daily CF from the Moderate Resolution Imaging Spectroradiometer (MODIS) onboard Terra, PA from the Global Precipitation Measurement (GPM) Integrated Multi-satellite Retrievals for GPM (IMERG), and photosynthetically active radiation (PAR) from the Clouds and the Earth's Radiant Energy System (CERES). The cross-hatched areas represent regions where the P-value from a Student's t-test is less than 0.001.

Changes in Manuscript:

“In addition, the direct correlation analysis shows statistically strong positive correlations between PA* and CF* and strong negative correlations between PAR* and CF* at the global scale (Supplementary Fig. 14), further supporting the use of total CF as an effective combined proxy for precipitation and radiation in this study.” [Lines 394–398 in the “Track Changes” version]

Comment 2.2:

Secondly, FLUXCOM-X-BASE shows a positive sensitivity of GPP to cloud fraction in evergreen broadleaf forests, which contradicts the results from FLUXCOM-RS. The authors attribute this discrepancy to uncertainties in the representation of water processes within the FLUXCOM framework. This explanation remains somewhat vague; a more in-depth exploration of the possible causes is recommended. For example, does the use of meteorological drivers in X-BASE introduce additional collinearity? Or does the structure of its machine learning model respond differently to radiation—particularly diffuse light—compared to the remote sensing (RS) product?

Response:

We thank the reviewer for highlighting this issue and for suggesting potential explanations. In our study, we found differences in the GPP-to-CF sensitivity between FLUXCOM-X-BASE and FLUXCOM-RS at two time scales: (1) at the 8-daily scale, X-BASE shows a globally higher GPP*-to-CF* sensitivity compared to RS (Supplementary Fig. 22); (2) at the annual scale, X-BASE exhibits positive GPP-to-CF sensitivity in evergreen broadleaf forests, whereas RS shows negative sensitivity (Supplementary Fig. 25). These differences, in the revised version, are interpreted in light of known FLUXCOM limitations documented in Nelson et al. (2024), particularly: (i) the insufficient representation of water-related effects and (ii) limited skill in capturing interannual and decadal variability.

Firstly, at the 8-daily scale, the higher global GPP*-to-CF* sensitivity in X-BASE is likely related to water-related uncertainties. X-BASE systematically shows higher evapotranspiration (ET) than RS (Nelson et al., 2024, Fig. 8), and its agreement with sun-induced fluorescence (SIF) from the Sentinel-5P TROPOMI instrument is weaker when compared to RS (Nelson et al., 2024, Fig. 7). This decrease in performance indicates the uncertainty related to the predictor variable set for capturing water-related effects (Nelson et al., 2024). For this evidence, we use the uncertainties in the representation of water processes for the explanation of the 8-daily scale issues.

Secondly, at the annual scale, the positive GPP-to-CF sensitivity in evergreen broadleaf forests from X-BASE, compared to the negative sensitivity in RS, can be partly attributed to the uncertainty of interannual variability. X-BASE GPP generally shows weaker interannual variability than RS (0.575 vs.

1.023 Pg C yr⁻¹; Nelson et al., 2024, Table C2), which may contribute to differences in annual-scale GPP-to-CF sensitivity.

Regarding the potential explanations proposed by the reviewer, we note that while the additional use of meteorological drivers in X-BASE could, in principle, introduce collinearity, we did not find explicit evidence for this and therefore cannot draw a definitive conclusion. Similarly, concerning whether the structure of the X-BASE machine learning model responds differently to radiation, particularly diffuse versus direct components, X-BASE only uses total incoming shortwave radiation as input without distinguishing between diffuse and direct fractions, so we cannot make an inference on this aspect either. Nevertheless, we sincerely appreciate the reviewer for raising these open-ended questions, which provide valuable avenues for future exploration.

Overall, we have clarified these points in the revision. Note that we have moved the section “comparison with independent datasets” to the Supplementary Information as suggested by review #1 to help readers follow the main narrative without being distracted by excessive detail.

Reference:

Nelson, J. A. et al. X-BASE: the first terrestrial carbon and water flux products from an extended data-driven scaling framework, FLUXCOM-X. *Biogeosciences* **21**, 5079-5115, doi:10.5194/bg-21-5079-2024 (2024).

Changes in Manuscript:

For the 8-daily scale:

“This difference may partly reflect the limited representation and associated uncertainties of capturing water-related processes in the FLUXCOM framework. As documented by ref.¹, FLUXCOM-X-BASE systematically shows higher evapotranspiration (ET) than FLUXCOM-RS, and its agreement with sun-induced fluorescence (SIF) from the Sentinel-5P TROPOMI instrument is weaker when compared to RS.” [Pages 2–3 in the Supplementary Information]

For the annual scale:

“This difference can partly be attributed to uncertainties in interannual variability. As documented by ref.¹, FLUXCOM-X-BASE exhibits weaker interannual variability in GPP compared to FLUXCOM-RS (0.575 vs. 1.023 Pg C yr⁻¹), which may contribute to the differences in annual-scale GPP-to-CF sensitivity.” [Page 4 in the Supplementary Information]

Reference:

1 Nelson, J. A. et al. X-BASE: the first terrestrial carbon and water flux products from an extended data-

driven scaling framework, FLUXCOM-X. *Biogeosciences* **21**, 5079-5115, doi:10.5194/bg-21-5079-2024 (2024).

Comment 2.3:

Thirdly, the argument that CF is an effective proxy for the combination of PAR and PA is robust on a global, climatic scale. However, the manuscript could more openly acknowledge a key limitation: at the instantaneous or daily scale, the relationship between CF and precipitation is weak and non-linear. Similarly, the conclusion about a spatial shift in GPP is based on a robust multi-dataset consensus on declining CF. However, the projection figure (Fig. 4) relies on the annual-scale sensitivity (β_{CF}) derived from the recent past (2001-2020) remaining constant throughout the 21st century. This is a strong assumption. The authors should explicitly state this as a key uncertainty in the discussion. Could β_{CF} itself change with increasing CO_2 , VPD, or vegetation acclimation? A brief discussion on this would improve the paper's balance.

Response:

We thank the reviewer for pointing this out. These key limitations are further discussed in detail in the revised version, as suggested by the reviewer.

Changes in Manuscript:

“However, our statistical investigation of the CF^* – PA^* relationship is based on 8-daily averages at the global scale. At instantaneous or daily timescales, this relationship is expected to be weaker and more non-linear⁷⁴, which should be taken into account when interpreting short-term dynamics.” [Lines 398–401 in the “Track Changes” version]

“It should be noted that clouds remain one of the most uncertain components in climate models because their sub-grid processes are not explicitly resolved and rely on imperfect parameterizations^{69,70}. These limitations may contribute to uncertainties in projected CF changes and, consequently, in the estimated vegetation responses. Moreover, under a warming climate, future changes in clouds, radiation, precipitation, and vegetation are expected to coevolve with adjustments in the hydrological cycle. Here, we highlight several potential processes that respond to climate change: (i) the atmosphere’s water-holding capacity increases by about $7\% \text{ }^\circ\text{C}^{-1}$, while global precipitation rises more slowly by roughly $2\text{--}4\% \text{ }^\circ\text{C}^{-1}$ due to energetic constraints⁷¹; (ii) cloud vertical structure is projected to change, featuring a faster decline in liquid clouds than in ice clouds over land, thereby influencing cloud radiative effects³³; (iii) GPP responds negatively to increasing VPD while exhibiting a nonlinear response to soil moisture¹⁸; (iv) thermal acclimation buffers photosynthesis against warming, reducing increases in photosynthesis under cold climates and limiting declines in photosynthesis under hot climates⁷². Collectively, these adjustments imply

that cloud and vegetation characteristics will vary according to regional hydroclimate, potentially altering the radiative and precipitative impacts on GPP and introducing uncertainties in the estimation of CF-driven GPP changes shown in Fig. 4. Therefore, as our projections assume that the GPP-to-CF sensitivity inferred from the recent past (2001–2020) remains constant throughout the 21st century, future studies should aim to quantify the potential uncertainties under changing CO₂, VPD, hydrological cycle, and vegetation acclimation.” [Lines 288–306 in the “Track Changes” version]

References:

- 18 Fu, Z. *et al.* Critical soil moisture thresholds of plant water stress in terrestrial ecosystems. *Science Advances* **8**, eabq7827, doi:10.1126/sciadv.abq7827 (2022).
- 33 Luo, H., Quaas, J. & Han, Y. Examining cloud vertical structure and radiative effects from satellite retrievals and evaluation of CMIP6 scenarios. *Atmospheric Chemistry and Physics* **23**, 8169-8186, doi:10.5194/acp-23-8169-2023 (2023).
- 69 Konsta, D. *et al.* Low-level marine tropical clouds in six CMIP6 models are too few, too bright but also too compact and too homogeneous. *Geophysical Research Letters* **49**, e2021GL097593, doi:10.1029/2021GL097593 (2022).
- 70 Klein, S. A. *et al.* Are climate model simulations of clouds improving? An evaluation using the ISCCP simulator. *Journal of Geophysical Research: Atmospheres* **118**, 1329-1342, doi:10.1002/jgrd.50141 (2013).
- 71 Held, I. M. & Soden, B. J. Robust responses of the hydrological cycle to global warming. *Journal of Climate* **19**, 5686-5699, doi:10.1175/JCLI3990.1 (2006).
- 72 Schneider, P. D., Gessler, A. & Stocker, B. D. Global photosynthesis acclimates to rising temperatures through predictable changes in photosynthetic capacities, enzyme kinetics, and stomatal sensitivity. *Journal of Advances in Modeling Earth Systems* **17**, e2024MS004789, doi:10.1029/2024MS004789 (2025).
- 74 Jin, D., Oreopoulos, L., Lee, D., Tan, J. & Cho, N. Cloud–precipitation hybrid regimes and their projection onto IMERG precipitation data. *Journal of Applied Meteorology and Climatology* **60**, 733-748, doi:10.1175/JAMC-D-20-0253.1 (2021).

Comment 2.4:

Finally, given that the Earth has undergone global dimming and brightening—a phenomenon closely linked to changes in cloud cover and aerosols—how consistent are the paper’s results with these large-scale trends? It is advised to include relevant discussion on this point.

Response:

The global dimming and brightening phenomena are indeed relevant to our findings. Our analysis is based on CF observations during the brightening period, which show a decline in CF consistent with increased surface solar radiation. However, CF trends are not monotonic across all regions, and the observed brightening signal also reflects the reduction in aerosol levels in many regions. We have added a corresponding discussion in the revised manuscript.

Changes in Manuscript:

“Besides, historical records show that the Earth has undergone periods of global dimming (1950s–1980s) and brightening (since the late 1980s), primarily driven by variations in clouds and aerosols^{67,68}. Our analysis, based on observations during the brightening period, is consistent with this large-scale trend, as the observed decline in global average CF corresponds to reduced cloud albedo and increased surface solar radiation. However, CF trends are not monotonic across all regions, and the observed brightening signal also reflects the reduction in aerosol levels in many regions⁶⁶.” [Lines 262–268 in the “Track Changes” version]

References:

- 66 Quaas, J. et al. Robust evidence for reversal of the trend in aerosol effective climate forcing. *Atmospheric Chemistry and Physics* 22, 12221-12239, doi:10.5194/acp-22-12221-2022 (2022).
- 67 Wild, M. Global dimming and brightening: A review. *Journal of Geophysical Research: Atmospheres* 114, doi:10.1029/2008JD011470 (2009).
- 68 Wild, M. et al. Global dimming and brightening: An update beyond 2000. *Journal of Geophysical Research: Atmospheres* 114, doi:10.1029/2008JD011382 (2009).

A few specific comments are given below.

The numbers in front of the comments indicate line number.

Comment 2.5:

1. L121-125. The paragraph describing the TRENDY models could be clearer. It correctly states that these models do not use cloud as a direct forcing factor, yet the analysis correlates their GPP output with MODIS cloud fraction (CF). While this is a valid methodological approach, the text should be rephrased to avoid any potential misunderstanding that cloud data is used as an input to the models themselves.

Response:

A very valuable suggestion by the reviewer. We have now rephrased this paragraph.

Changes in Manuscript:

“The GPP*-to-CF* sensitivity estimated using TRENDY modelled GPP and MODIS/Terra observed CF exhibits spatial patterns similar to those observational results discussed above, with sensitivity decreasing as HI increases (Supplementary Fig. 2). The MODIS/Terra CF is used only for diagnostic correlation with TRENDY GPP, not as a model input.” [Lines 124–128 in the “Track Changes” version]

Comment 2.6:

2. L194. The term ‘grid mean COT’ is ambiguous. It should be explicitly defined—for example, as “grid-mean cloud optical thickness” (τ), where $\tau = CF \times COT_avg$, and COT_avg represents the average cloud optical thickness over cloudy pixels only.

Response:

Well spotted and corrected now. We now define the grid-mean cloud optical thickness (\overline{COT}) as $\overline{COT} = CF \times \overline{COT}_{cloudy}$, where \overline{COT}_{cloudy} represents the average cloud optical thickness over cloudy pixels only.

Changes in Manuscript:

“Moreover, the grid-mean COT (\overline{COT}), defined as $\overline{COT} = CF \times \overline{COT}_{cloudy}$, where \overline{COT}_{cloudy} represents the average COT over cloudy pixels only, is applied to study the sensitivity of GPP to cloud vertical extension.” [Lines 199–201 in the “Track Changes” version]

“ \overline{COT}_{cloudy} represents the average optical thickness within cloudy areas, while multiplying it by CF yields the grid-mean COT (\overline{COT}), accounting for both cloudy and cloud-free regions (i.e., $\overline{COT} = CF \times \overline{COT}_{cloudy}$).” [Lines 559–561 in the “Track Changes” version]

Comment 2.7:

3. L495. The use of VODndr, which has relatively weak correlation with GPP (as shown in Suppl. Fig. 14f), calls for a more cautious interpretation of the resulting findings.

Response:

We thank the reviewer for pointing this out. We acknowledge that VODndr shows a relatively weak correlation with GPP compared to other photosynthetic proxies. We included this dataset to obtain additional, independent information, as VODndr is the only indicator derived from an active remote sensing method. As suggested, we have more clearly noted its limitations and advise caution when interpreting the results.

Changes in Manuscript:

“Further validation of VODndr against FLUXNET GPP supports this inference (Supplementary Fig. 16f), while we acknowledge the relatively weak correlation compared to other indicators, reflecting the inherent limitations of this proxy. Nevertheless, VODndr provides complementary information derived from an independent, active remote sensing approach, which is not captured by other datasets. By including VODndr alongside other indicators, we aim to enhance the robustness of our findings while being careful not to over-interpret its specific contributions.” [Lines 534–540 in the “Track Changes” version]

Comment 2.8:

4. L1064. Legend in panel a is unclear. Do the values 0.1, -0.1, and -0.3 correspond to all sites collectively? It is also difficult to distinguish between the representations of -0.1 and -0.3. Clarification is needed.

Response:

Thank you! We have adjusted the legend as suggested (Fig. R2.2). We also clarified in the figure caption that the values 0.1, -0.1, and -0.3 (now -0.5) in the legend serve only as a reference scale and do not correspond to individual site values. Moreover, the legend has been further optimized to display 0.1, -0.1 and -0.5, providing a clearer representation of the differences in triangle sizes. Other figures with the same issue have also been adjusted (Extended Data Fig. 1 and Fig. 2, Supplementary Fig. 4 and Fig. 6).

Figure R2.2. Humidity index (HI) spatially shapes the sensitivity of gross primary productivity (GPP) to cloud fraction (CF) across global ecosystems. (a) GPP^{*}-to-CF^{*} sensitivity (equation (4), Methods) derived from daily GPP measurements at FLUXNET and CF matched from the nearest grid of European Centre for Medium-Range Weather Forecasts (ECMWF) fifth generation reanalysis (ERA5) data. The size of the triangles represents the magnitude of the values, with upward triangles indicating positive values and downward triangles indicating negative values. **The triangles in the legend serve as reference scales for the values of 0.1, -0.1, and -0.5.** The colour filling inside each triangle corresponds to the average HI observed during the measurement period at each site. (b) Scatter plot showing the variation in GPP^{*}-to-CF^{*} sensitivity as a function of HI across sites in (a) on a logarithmic scale. GPP^{*}-to-CF^{*} sensitivity is modelled as a linear regression of the logarithmic HI, with the texts displaying the slope of the linear fit, correlation coefficient (*R*), and *P*-value from a Student's *t* test. The linear regression is represented by the black line. The size of each scatter point represents the length of the data record at each site, with larger points indicating longer measurement periods. Different colours are used to distinguish between different ranges of measurement durations. (c) Map of GPP^{*}-to-CF^{*} sensitivity derived from 8-daily GPP from FLUXCOM-RS and CF from the Moderate Resolution Imaging Spectroradiometer (MODIS) onboard Terra. Non-vegetated areas are masked (Methods). The cross-hatched areas represent regions where the *P*-value from a Student's *t*-test is less than 0.001. (d) As in (b) but with density plot across grids in (c), and HI calculated based on data from the Climatic Research Unit Time-Series Version 4.08 (CRU TS v4.08). The colour bar shows the Gaussian kernel density estimate. The analysis in (c) and (d) is based on the periods between 2001 and 2020.

Responses to comments on “Hydroclimate shapes photosynthetic sensitivity to cloud cover across global ecosystems” (NCOMMS-25-69717A)

Hao Luo^{1*}, Ana Bastos², Markus Reichstein³, Gregory Duveiller³,
Jan Kretzschmar¹, Johannes Quaas^{1,4}

¹Leipzig Institute for Meteorology, Leipzig University, 04103, Leipzig, Germany

²Institute for Earth System Science and Remote Sensing, Leipzig University, 04103, Leipzig, Germany

³Max Planck Institute for Biogeochemistry, 07745, Jena, Germany

⁴German Centre for Integrative Biodiversity Research (iDiv) Halle-Jena-Leipzig, 04103, Leipzig, Germany

*Corresponding author. Email: hao.luo@uni-leipzig.de

We are grateful to the two reviewers for their positive, thoughtful, and constructive comments, which are very helpful in improving the quality of this manuscript. We have thoroughly revised the manuscript to address all comments with point-by-point responses. We hope our revised version will receive favourable consideration and meet the journal’s requirements. The reviewers’ comments are reproduced (*black, italic*) along with our replies (*blue*) and changes made to the text (*red*). All the authors have read the revised manuscript and agreed with the submission in its current form.

Responses to Reviewer #2

Reviewer #2 (Remarks to the Author):

The authors have engaged substantively with every comment, and the revised manuscript reflects their careful attention to detail. I believe it would make an excellent contribution to Nature Communications.

Response:

We appreciate the reviewer's satisfaction with our revisions and are grateful for the important contributions to this study.

Responses to Reviewer #3

Reviewer #3 (Remarks to the Author):

The authors have addressed previous critiques with substantial revisions, and I recommend publication after the following important corrections are made.

Response:

We would like to thank the reviewer for taking the time to review this manuscript again and for providing valuable comments and constructive suggestions.

Comment 3.1:

1. Title. The manuscript focuses exclusively on terrestrial vegetated areas and excludes oceanic ecosystems involving photosynthesis. Therefore, the term "global ecosystems" is misleading and should be clarified, for example, as "global terrestrial ecosystems."

Response:

Excellent help by the reviewer! The title has been changed to “Hydroclimate shapes photosynthetic sensitivity to cloud cover across global terrestrial ecosystems”.

Changes in Manuscript:

“Hydroclimate shapes photosynthetic sensitivity to cloud cover across global terrestrial ecosystems”
[Lines 1–2 in the “Track Changes” version and Page 1 in the Supplementary Information]

Comment 3.2:

2. Abstract. It is unclear how long the datasets cover that support the finding that "the sensitivity of photosynthesis to cloud cover is spatially shaped by the hydroclimate." Additionally, in the conclusion, I recommend clearly stating that the further examination is based on "CMIP6 projections under the SSP585 scenario (2015–2099)." Please also change "GPP is expected to decline" to "GPP is projected to decline."

Response:

This is a valuable suggestion which we have followed. The FLUXNET measurements span 1994–2014, but the temporal coverage differs among individual FLUXNET sites. The global gridded datasets used to calculate the sensitivity cover the period 2001–2020. To avoid ambiguity and redundancy, we have added a concise description (“spanning the past few decades”) in the Abstract, while the detailed temporal coverage of each dataset is provided in the Methods section. In addition, we have clarified that the future assessment is based on CMIP6 projections under the SSP585 scenario (2015–2099). We have also revised the wording

from “GPP is expected to decline” to “GPP is projected to decline”.

Changes in Manuscript:

“Here, using observational- and model-based datasets spanning the past few decades, we show that...”
[Lines 16–17 in the “Track Changes” version]

“The findings indicate that, under a warming climate, particularly in the Coupled Model Intercomparison Project Phase 6 “ssp585” scenario (2015–2099), GPP is projected to decline in arid regions and increase in humid regions due to changes in cloud cover, suggesting an exacerbation of regional disparities in ecosystem functions.” [Lines 25–28 in the “Track Changes” version]

“Finally, based on the annual scale spatially resolved sensitivity metric of GPP to CF, we find for a warming climate, particularly in the CMIP6 “ssp585” scenario (2015–2099), GPP is projected to decrease in arid regions and increase in humid regions due to the decline in global average CF.” [Lines 370–373 in the “Track Changes” version]

Comment 3.3:

3. Fig. 4. The changes are presented relative to the period 1984–1988. Could the authors explain why this specific period was chosen instead of the longer “historical” experiment period (e.g., 1984–2014)?

Response:

We thank the reviewer for this question. We selected the 1984–1988 mean as the reference baseline in Fig. 4 to set the initial values of the three lines close to zero, which facilitates clearer interpretation of subsequent changes over time. This approach is intended to help readers intuitively grasp the magnitude and direction of changes relative to a common starting point. We believe that our conclusions remain the same regardless of the choice of reference values. However, if the 1984–2014 mean is used as the reference, the initial values of different lines can vary substantially; for example, in Fig. 4d, the blue line would start from a very negative value, while the orange line would begin at a very positive value, potentially complicating straightforward interpretation. We explain this now in the figure caption.

Changes in Manuscript:

“Estimated changes in GPP caused by changes in CF relative to the beginning of the depicted period (1984–1988 mean) (equation (11), Methods) using...” [Lines 1141–1143 in the “Track Changes” version]

Comment 3.4:

4. Line 118. Please replace “S2” with a clearly defined term, such as “the high-emission pathway (SSP5-8.5).”

Response:

We thank the reviewer for pointing this out. Dynamic Global Vegetation Models (DGVMs) participated in the “Trends and drivers of the regional scale terrestrial sources and sinks of carbon dioxide” (TRENDY) project perform four factorial simulations (Sitch et al., 2024):

- S0: A control pre-industrial simulation in which all forcings stay as in the spin-up. S0 allows to diagnose any “cold start” issues and model drift.
- S1: In this simulation observed global CO₂ evolves over the historical period, the climate and land cover forcings stay at their pre-industrial levels like in simulation S0.
- S2: In this simulation, CO₂ and climate evolve over the historical period, while the land cover stays at its pre-industrial level.
- S3: All forcings (CO₂, climate and land use) evolve over the historical period in this simulation. The meteorological data is used in the same way as in the S2 simulation.

The term “S2” is the official designation in the TRENDY project for the simulation with evolving climate and CO₂ but fixed land cover, and this has been described in the original manuscript “Our analysis uses outputs from simulation scenario S2 (Methods), which incorporates temporal variations in climate and CO₂ concentrations, while keeping land cover stable at preindustrial levels to exclude the potential confounding effects from land use changes.” [Lines 117–121 in the “Track Changes” version]. This point has now been clarified more clearly in the revised version.

Reference:

Sitch, S. et al. Trends and drivers of terrestrial sources and sinks of carbon dioxide: An overview of the TRENDY project. *Global Biogeochemical Cycles* **38**, e2024GB008102, doi:10.1029/2024GB008102 (2024).

Changes in Manuscript:

“Our analysis uses outputs from simulation scenario S2 (scenario defined in the TRENDY project, more details in the Methods section), which...” [Lines 117–119 in the “Track Changes” version]

Comment 3.5:

5. Line 769. The longest lag l is set to -8 . How was the maximum lag time of 64 days determined? Please clarify the rationale for this choice.

Response:

We thank the reviewer for this constructive suggestion. Here, we performed quantitative analyses (Table R3.1) to justify the choice of longest lag $l = -8$. Specifically, we calculated the proportion of samples

for each effective response period (τ) when $l = -8$ and found that for $\tau \leq -7$, fewer than 3% of samples contribute, indicating a negligible contribution from longer lags. Furthermore, sensitivity analyses with longest lags of $l = -9$ and $l = -10$ have similar results, with the proportion of samples for $\tau \leq -7$ remaining below 3% globally. Results from the $l = -9$ and $l = -10$ cases also suggest that nearly 99% of pixels globally exhibit delayed effects shorter than two months ($\tau \geq -8$). These quantitative analyses support the use of $l = -8$ (64 days) as an appropriate cutoff that captures the substantial lagged effects while excluding negligible contributions.

Table R3.1. Sample proportions of effective response period (τ) of gross primary productivity (GPP) to precipitation amount (PA). Only pixels with a statistically significant sensitivity (P -value < 0.001) are included. Three sensitivity analyses with longest lag $l = -8, -9, -10$ are shown.

	$\tau = 0$	$\tau = -1$	$\tau = -2$	$\tau = -3$	$\tau = -4$	$\tau = -5$	$\tau = -6$	$\tau = -7$	$\tau = -8$	$\tau = -9$	$\tau = -10$
$l = -8$	44.5%	15.3%	16.6%	10.2%	6.2%	3.1%	1.6%	1.1%	1.4%	–	–
$l = -9$	44.2%	15.2%	16.3%	10.2%	6.3%	3.2%	1.7%	1.1%	0.7%	1.1%	–
$l = -10$	44.8%	14.8%	16.2%	10.1%	6.2%	3.2%	1.7%	0.9%	0.7%	0.6%	0.6%

Changes in Manuscript:

“We also test longest lag $l = -9$ and $l = -10$. Both cases suggest that nearly 99% of pixels globally exhibit delayed effects shorter than two months ($\tau \geq -8$). Therefore, the use of $l = -8$ is appropriate, as it captures the substantial lagged effects while excluding negligible contributions.” [Lines 711–713 in the “Track Changes” version]